# Neural dissociation of attention and working memory through inhibitory control

Yueyao Liu[1], Yingtao Fu[1], Enze Tang [1], Hang Wu[2], Junrong Han[2], Musi Xie[2], Yihui Zhang[2], Bo Peng[3], Jinhao Huang[2], Hanjun Liu [3], Hui Chen [1,4,5] ✉ & Pengmin Qin [2,6,7] ✉

Attention and working memory (WM) have traditionally been considered closely linked processes with shared neural mechanisms. In information selection, attention is often conceptualized as a gatekeeper to WM, regulating which information is encoded and stored. Here, combining tasks specifically designed to separate attention from WM encoding with a multimodal approach, we provide converging neural and causal evidence that these processes are dissociable. Functional MRI identifies the supramarginal gyrus (SMG) as the key region enabling this dissociation, while dynamic causal modeling reveals the neural circuitry through which the SMG exerts inhibitory control over attentional representations, regulating their integration into WM. Furthermore, neuromodulation via transcranial direct current stimulation (tDCS) demonstrates that enhancing SMG activity strengthens this inhibitory control. A second tDCS experiment using varied stimuli confirms the generalizability of the effect. Finally, a transcranial magnetic stimulation (TMS) experiment provides further causal evidence with greater spatial precision. These findings challenge the long-standing view that attention and WM encoding form a continuous process, demonstrating instead that they constitute two dissociable neural processes of information selection.

Attention and working memory (WM) are fundamental cognitive functions that support goal-directed behavior[1]. Attention enables individuals to focus on relevant information in complex environments, while WM temporarily stores selected inputs for cognitive processing. Understanding their interaction is critical for uncovering how the brain prioritizes, processes, and retains information. Traditionally, attention and WM have been considered tightly linked in their neural underpinnings. Neuroimaging and electrophysiological studies in humans and macaques suggest that attentional selection and WM processes share a common neural basis, primarily in the prefrontal and parietal cortices[2–7].

However, recent evidence suggests that attention and WM may rely on distinct neural mechanisms. Mendoza-Halliday et al. (2024) trained macaques to either maintain features in WM or use them to guide attentional deployment, identifying separate neuronal substrates in the posterior lateral prefrontal cortex for WM storage and feature-based attention[8]. This finding highlights the dissociation between maintaining WM representations and using them to guide attentional allocation. However, traditional models emphasizing the close interdependence between attention and WM have primarily focused on another aspect—the relationship between attentional selection and WM encoding. From the traditional perspective,

[1]Department of Psychology and Behavioral Sciences, Zhejiang University, Hangzhou, China. [2]School of Psychology, South China Normal University, Guangzhou, China. [3]Department of Rehabilitation Medicine, The First Affiliated Hospital of Sun Yat-sen University, Guangzhou, China. [4]The State Key Lab of Brain-Machine Intelligence, Zhejiang University, Hangzhou, China. [5]Zhejiang Key Laboratory of Neurocognitive Development and Mental Health, Zhejiang University, Hangzhou, China. [6]Guangdong Key Laboratory of Mental Health and Cognitive Science, Guangzhou, China. [7]Key Laboratory of Brain, Cognition and Education Sciences, Ministry of Education, Guangzhou, China. ✉e-mail: chenhui@zju.edu.cn; qin.pengmin@m.scnu.edu.cn

attentional filtering is seen as the gatekeeper of WM encoding for external inputs, with the assumption that attended information is automatically encoded into WM[3,9–11]. Consistent with this view, prior work has demonstrated the neural basis for such a gatekeeping function (e.g., activity in the prefrontal cortex and basal ganglia preceding the filtering of irrelevant information[7]). However, in conventional experimental designs, attentional selection and WM encoding remain difficult to dissociate, as attended information is typically required to be encoded into WM[5,6,11].

Here, we introduce a design to address this limitation, enabling a direct dissociation between attentional selection and WM encoding within a single behavioral task[12,13]. Unlike traditional paradigms, the task requires participants to attend to a piece of information without reporting it later (see Methods for task details), thereby separating attentional selection from WM encoding. This method provides a unique opportunity to examine the neural mechanisms underlying their dissociation. Understanding this neural separation is crucial, as it challenges the long-standing assumption that attention and WM constitute a continuous process rather than two distinct stages of information selection.

Based on this design, we employ a multimodal approach integrating neuroimaging, computational modeling, and neurostimulation. Using functional MRI (fMRI), we identify brain regions involved in regulating the transition of attended representations into WM, distinguishing those that facilitate this transfer from those that suppress unnecessary encoding. To further investigate the dynamic neural interactions governing this process, we construct computational models of neural circuits to examine how different brain regions coordinate WM integration for attended representations. Moreover, we apply transcranial direct current stimulation (tDCS) to modulate activity in key regions identified through fMRI and computational modeling, providing causal evidence for how neural activity influences the gating of representations from attention to WM. To ensure the robustness and generalizability of our findings, we replicate the tDCS experiments across diverse stimulus types. Finally, we employ transcranial magnetic stimulation (TMS) to provide additional causal evidence with greater spatial specificity. Uncovering this dissociable neural basis redefines our understanding of how the brain prioritizes and organizes information to support goal-directed cognition.

## Results
### Attended information can be blocked from working memory
To investigate the neural mechanisms underlying the dissociation between attentional selection and WM encoding, we conducted an fMRI experiment using a design modified from the attribute amnesia paradigm[12] to isolate these two processes. As illustrated in Fig. 1, participants were tested under two experimental conditions: the pre-surprise condition, in which the face was attended without memory expectation, and the post-surprise condition, in which participants expected to report the face and thus encoded it into working memory. While participants performed the same face location-reporting task in both conditions, the memory relevance of the face varied. Consequently, the same face stimuli served either as attended-without-memory or attended-with-memory representations, providing a

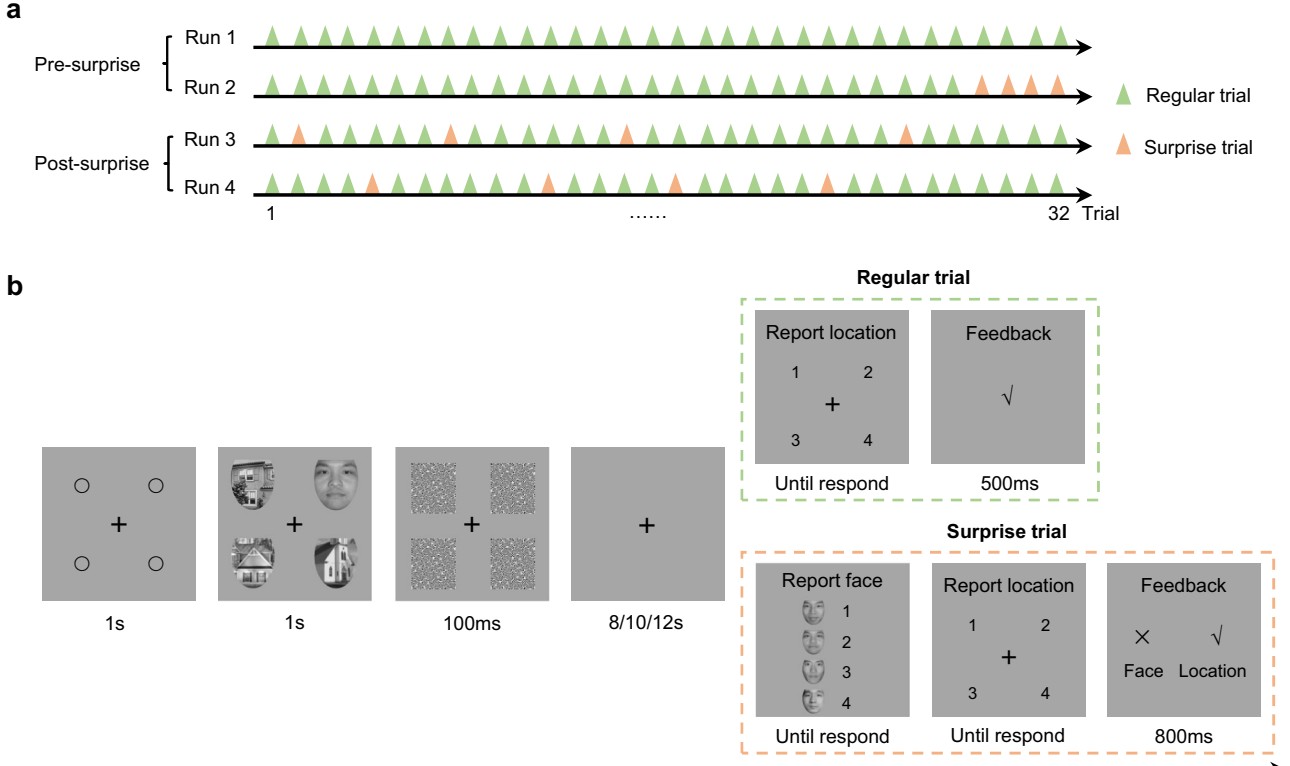

**Fig. 1 | fMRI experimental design and task procedure. a** The attribute amnesia experiment comprised four runs, each containing 32 trials. In the pre-surprise condition, participants completed 60 regular trials (green triangles), performing a face location-reporting task that required visual search within a stimulus array. Although the face was fully attended and processed within a few hundred milliseconds[15–19], participants were instructed to remember only its location. In the post-surprise condition (56 regular trials), participants continued the same location task, but now anticipated identity reports. Surprise trials served as catch trials to monitor face processing and sustain the expectation of face reporting. **b** In regular trials, participants are instructed to report the location of the face. In surprise trials, participants are instructed to select the target faces and then report the location. The instruction for the location-reporting task is "Press a corresponding number to indicate the location of the face." The instruction for the face-reporting task is "This is a surprise memory test! Press a corresponding number to indicate which one is the face you just saw." Example stimuli from the Chinese Affective Face Picture System[76], shown for illustration only and used with permission.

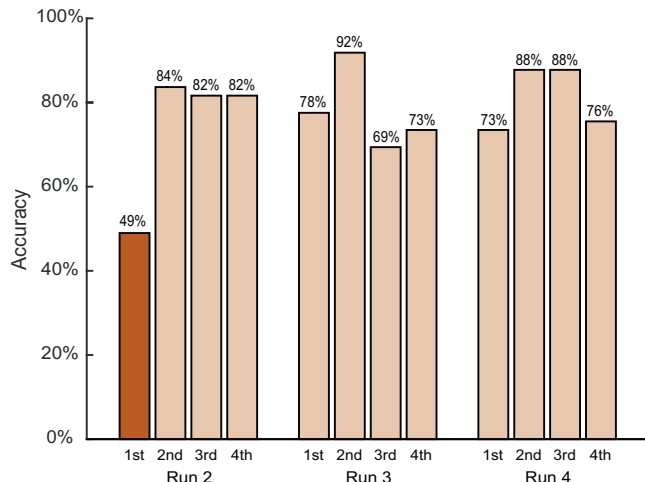

**Fig. 2 | Behavioral results in the face-reporting task across surprise trials.**
Accuracy in the first surprise trial of Run 2 (dark orange bar, 49%, 24 out of 49 participants) was significantly lower than in all subsequent control trials (light orange bars, all $ps < 0.05$, two-sided chi-squared). This difference remained significant even when compared specifically with the least accurate control trial (69%, 34 out of 49 participants), 49% vs. 69%, $\chi^2(1, N = 49) = 4.224$, $p = 0.040$, $\varphi = 0.294$. Consistent with previous reports of the attribute amnesia effect[12,20,87,88], participants failed to report attended information during the first unexpected probe, but performance improved rapidly once memory demands were anticipated. Accuracies of each trial are displayed as correct/total participants, rounded to integers. Source data are provided as a Source Data file.

unique opportunity to dissociate attention and working memory encoding within the same task context.

According to the behavioral results, participants maintained high accuracy in the location task, with 97.64% accuracy. This indicates that they accurately reported the target location and remained fully engaged in the experiment. As shown in Fig. 2, the accuracy of the face-reporting task in the first surprise trial was 48.98% (24/49), significantly lower than in subsequent surprise trials (all $ps < 0.05$, Chi-squared test). These results indicate that participants struggled to report the target face when it was attended-without-memory information but showed significant improvement in subsequent trials when it became attended-with-memory information. These findings validate the effectiveness of this design in dissociating attention from WM.

**Stronger activation of the SMG in the pre-surprise condition**
We identified brain regions involved in regulating the transition of attended representations into WM, distinguishing those that facilitate this transfer from those that suppress unnecessary memory encoding. To do so, we conducted a whole-brain paired $t$-test comparing the pre-surprise and post-surprise conditions, with statistical maps thresholded at $p < 0.01$ (FDR-corrected at the voxel level) and a cluster extent threshold of >40 voxels. As shown in Fig. 3, the right supramarginal gyrus (SMG) exhibited significantly greater activation in the pre-surprise condition, suggesting its role in inhibiting the encoding of target faces that are irrelevant to memory. In contrast, the left dorsolateral prefrontal cortex (dlPFC), left supplementary motor area (SMA), and left ventromedial prefrontal cortex (vmPFC) showed greater activation in the post-surprise condition, suggesting their involvement in the active storage of target faces in memory.

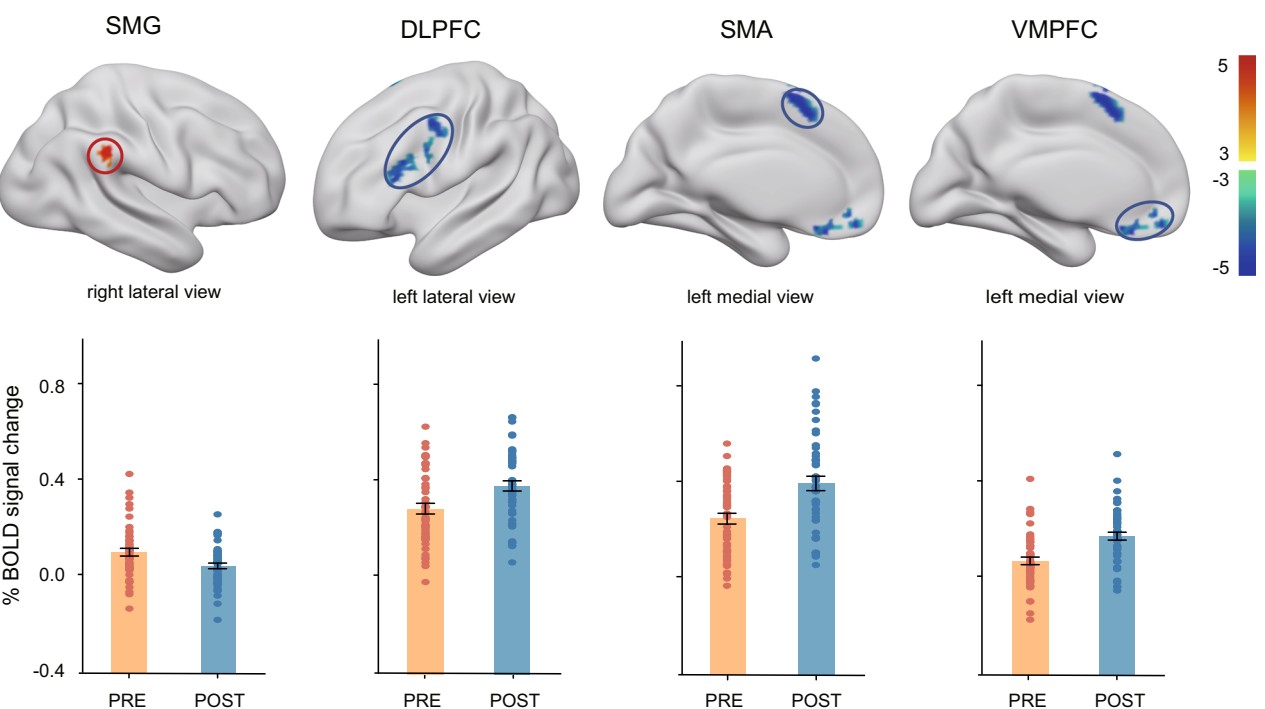

**Fig. 3 | Whole-brain analyses for the fMRI attribute amnesia task.** A general linear model (GLM) analysis was employed to compare whole-brain BOLD signals between the two conditions, with results shown for the contrast "pre-surprise > post-surprise". A paired t-test was applied with a significance threshold of $p < 0.01$, FDR correction at the voxel level, and a cluster size of >40 voxels. Positive activation (red) indicates stronger activation in the pre-surprise condition, whereas negative activation (blue) indicates higher activation in the post-surprise condition. The resulting clusters included the right supramarginal gyrus (SMG; 68 voxels, peak at [55, −44, 37]), left dorsolateral prefrontal cortex (dlPFC; 121 voxels, peak at [-44, 25, 22]), left supplementary motor area (SMA; 93 voxels, peak at [−2, 13, 58]), and left ventromedial prefrontal cortex (vmPFC; 62 voxels, peak at [−5, 25, −20]). Detailed statistical results are provided in the Supplementary Table 1. The bars depict the mean signals within the GLM-defined clusters for each condition, with colored dots showing individual participants; these summaries are descriptive only and not subjected to additional statistical tests, as inference is given by the voxel-wise FDR-corrected maps above. $n = 49$ participants. Source data are provided as a Source Data file.

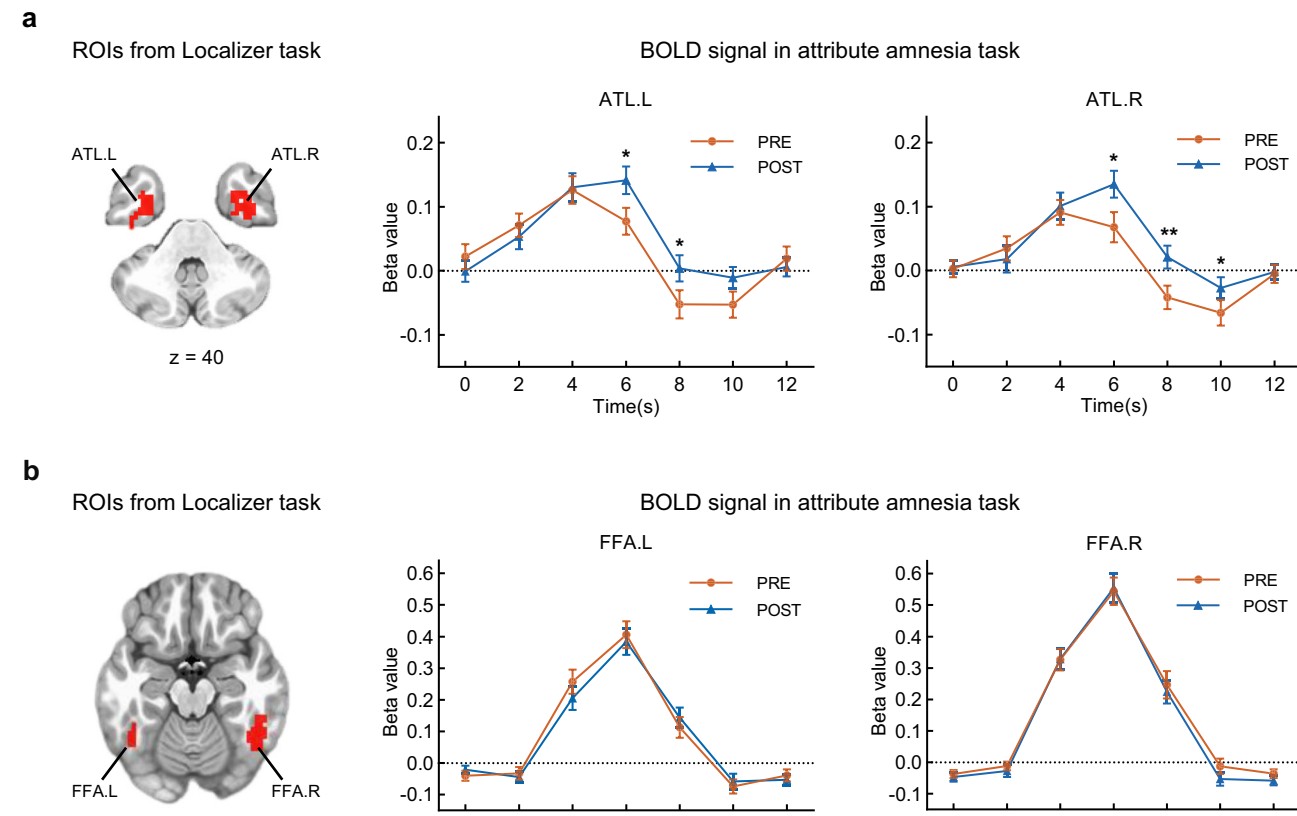

**Fig. 4 | ROI analysis for face perception. a** The regions of interest (ROIs) for the bilateral anterior temporal lobe (ATL.L and ATL.R) are shown on the left, with corresponding beta value curves on the right. The time courses illustrate BOLD signal changes in the pre-surprise and post-surprise conditions of the attribute amnesia task. Error bars represent SEM. $n = 49$; two-sided paired $t$-test, uncorrected for multiple comparisons. Significant differences between conditions were observed in the left ATL at 6 s ($p = 0.020$) and 8 s ($p = 0.040$), and in the right ATL at 6 s ($p = 0.014$), 8 s ($p = 0.003$), and 10 s ($p = 0.048$). **b** The ROIs for the bilateral fusiform face area (FFA.L and FFA.R) are shown on the left, with corresponding beta value curves on the right, following the same format as in (**a**). *$p < 0.05$, **$p < 0.01$. Source data are provided as a Source Data file.

In this analysis, the right SMG, which may be associated with inhibitory function, survived the FDR-corrected threshold. To explore potential bilateral involvement, we conducted an exploratory analysis using a more lenient threshold ($p < 0.005$, uncorrected), which revealed weaker activation in the left SMG (see Supplementary Fig. 2), although this did not survive statistical correction. As the current results align with prior evidence[14] indicating a right-hemispheric dominance of parietal regions in executive control and top-down inhibition, we focused all subsequent analyses in the main text (e.g., dynamic causal modeling and neuromodulation) on the right SMG, hereafter referred to simply as SMG. For consistency, we refer to all regions by their anatomical labels without specifying hemisphere (e.g., SMG, dlPFC, SMA, vmPFC).

### Neural representation of face perception when it is attended without or with memory

Based on previous studies of the automatic attentional advantages of face processing[15–19] and the attribute amnesia effect[12,20,21], both category and identity representations are expected to be processed during the target face-location task in both conditions. Consequently, we focused on two key perceptual processing regions as regions of interest (ROIs) to examine face perception in the brain: the anterior temporal lobe (ATL), responsible for processing face identity[22,23], and the fusiform face area (FFA), responsible for category representations of faces[24,25]. To localize these ROIs, participants completed a face-localizer task[26] during the fMRI scan. Based on combined anatomical and functional criteria[22], we successfully identified bilateral ATL and bilateral FFA (Fig. 4). ROIs were defined as suprathreshold clusters centered on peak voxels in MNI space (LPI orientation), using a threshold of $p < 0.01$ (FDR-corrected at the voxel level). Specifically, this included the left ATL (65 voxels centered at [−32, −11, −38]), right ATL (93 voxels at [31, −2, −50]), left FFA (41 voxels at [−47, −56, −23]), and right FFA (110 voxels at [46, −59, −20]). See the Methods section and Supplementary Table 1 for further details.

To investigate neural responses to face perception, we performed an ROI analysis to calculate beta values for the two conditions. In each ROI, the time courses of beta values for each condition are displayed on the right side of Fig. 4. In the left ATL, two beta values (at 6 s and 8 s) in the pre-surprise condition were lower than those in the post-surprise condition. A similar pattern was observed in the right ATL, where three beta values (at 6 s, 8 s, and 10 s) in the pre-surprise condition were lower than those in the post-surprise condition. These beta value curves indicates that, compared with the post-surprise condition, bilateral ATL exhibits lower peak activation and earlier decay in the pre-surprise condition. However, no differences in activation pattern were observed in the left and right FFA between the two conditions. The lower neural responses in the ATL during the pre-surprise condition suggest that identity representations might be inhibited, whereas category representations might not be.

### SMG inhibits the activation of ATL in the pre-surprise condition
To investigate the neural circuits underlying the dissociation between attention and WM—particularly how the brain suppresses attended representations that are unnecessary for memory—we applied

dynamic causal modeling (DCM) analysis[27–31] to examine effective connectivity among key brain regions. This approach enables us to validate the hypothesis that the SMG inhibits face perception to prevent its entry into working memory[32], while also considering an alternative argument suggesting that all attended information is initially encoded into working memory and subsequently removed if unnecessary from memory storage[3,33–35]. The DCM analysis focused on connectivity changes among three key regions identified in prior analyses (Figs. 3 and 4): the inhibition region (SMG), the face perception region (ATL or FFA), and the WM storage region (dlPFC)[36]. All ROIs were defined based on suprathreshold activation clusters from whole-brain group-level contrast maps, using a consistent statistical threshold (voxel-wise FDR-corrected $p < 0.01$).

In the first stage of the DCM analysis (Fig. 5a), we defined 15 different models among three regions to model the modulatory effect of the pre-surprise condition using the post-surprise condition as the baseline, and then fitted each model for each participant[37]. In the second stage, we compared 15 different models at the group level by calculating the exceedance probability of each model—defined as the probability that a given model outperforms all others in explaining the observed data[38]. We then selected the best model using random effects Bayesian Model Selection[28]. The results demonstrated that for both the ATL or FFA, the 15th model best explained the modulatory effect of the pre-surprise condition.

Specifically, for the model of ATL, the pre-surprise condition significantly increased the suppression from the SMG to the left ATL (mean values of 49 participants = −0.209, $t(48) = −2.462$, $p = 0.017$, one-sample $t$-test, Fig. 5b), as well as to the right ATL (mean values of 49 participants = −0.238, $t(48) = −2.199$, $p = 0.033$, one-sample $t$-test, Fig. 5c). No significant modulatory effect was found for the other connections in the selected 15th model. As Fig. 5 shows, the brain models indicate a significant inhibitory effect from the SMG to the ATL under the pre-surprise condition. Additionally, for the model of FFA, there was no significant modulatory effect on either the connectivity from the SMG to the left/right FFA or any other inter-regional connectivity (Supplementary Fig. 1). This aligns with the ROI analysis for face perception activation patterns, providing further evidence that neural representations of face identity are inhibited in the pre-surprise condition, while category representations remain unaffected.

## Anodal tDCS on the SMG strengthens inhibitory control over attended inputs

To provide causal validation for our neuroimaging findings, we conducted two tDCS experiments targeting the SMG—the region identified as the inhibitory hub that prevents unnecessary attentional information from entering working memory. High-definition tDCS (2 mA for 20 min) was applied to the SMG[39], and ninety participants in each experiment were randomly assigned to one of three groups: anodal, cathodal, or sham stimulation[40] (Figs. 6a, b; see Methods for details).

In the first tDCS experiment, we used a modified change-detection paradigm with colored faces as stimuli (Fig. 6c). Compared with the attribute amnesia task, which relies on a single surprise trial, the change-detection task provided sufficient within-subject trials, making it more sensitive for detecting stimulation effects in the neuromodulation experiments. In each trial, participants first judged the identity of a memory item but were instructed to memorize only its color. Thus, the face identity was fully attended yet irrelevant to the memory report, serving as attended-without-memory information. Reaction time (RT) was used to assess residual memory traces of face identity. According to prior findings[13], similar RTs across same-identity and change-identity condition reflect baseline inhibition. Faster RTs for same-identity trials indicate weaker inhibition (i.e., greater residual memory trace of identity), while reversed RTs—where same-identity trials produce slower responses—indicate stronger inhibitory suppression of the identity.

We first conducted a full 3 (Stimulation: Anodal, Cathodal, Sham) × 2 (Condition: Same-identity, Change-identity) mixed-design ANOVA. the analysis revealed a trend-level interaction between stimulation and condition ($F(2,87) = 2.980$, $p = 0.056$, $\eta p^2 = 0.064$), with no significant main effects of stimulation or condition. This trend suggested the possibility of polarity-dependent effects and that combining both active groups in a single analysis may have masked polarity-specific modulation effects[41,42]. Based on previous literature, the polarity-specific effects are typically examined by directly comparing each active group with sham[41,43,44]. Therefore, we conducted planned 2 (Group: Active vs. Sham) × 2 (Condition) ANOVAs separately for the anodal and cathodal groups. As shown in Fig. 6d, the comparison between anodal and sham groups revealed a significant interaction effect ($F(1,58) = 6.465$, $p = 0.014$, $\eta p^2 = 0.100$). Post-hoc paired $t$-tests demonstrated that in the anodal stimulation group, RTs were significantly longer for the same-identity condition compared to the change-identity condition (MD [mean difference of change - same] = −22.21 ms, 95% CI = [−38.87, −5.54], $t(29) = −2.725$, $p = 0.011$, Cohen's d = -0.498), indicating a stronger inhibitory effect on the attended face identity and a reduction in memory traces. Conversely, in the sham group, there was no significant difference in RT between the two conditions (MD = 3.87 ms, 95% CI = [−8.86, 16.60], $t(29) = 0.621$, $p = 0.539$, Cohen's d = 0.113), reflecting a baseline level of memory traces of the identity[13]. No significant interaction was observed between the cathodal and sham groups ($F(1,58) = 0.254$, $p = 0.616$, $\eta p^2 = 0.004$), and there was no significant difference in RT between the two conditions in cathodal group (MD = −1.73 ms, 95% CI= [−20.53, 17.07], $t(29) = −0.188$, $p = 0.852$, Cohen's d = −0.034), suggesting no discernible differences in the inhibitory effect following cathodal stimulation.

Moreover, to further validate our findings and extend their generalizability, we expanded our investigation of the dissociation mechanism to include simpler stimuli. In the second tDCS experiment, we replicated the procedure from the first experiment, replacing face stimuli with colored shapes. Ninety new participants performed this modified change-detection task, with the shape identity serving as the attended-without-memory information (Fig. 7a).

We first conducted a 3 (Stimulation: Anodal, Cathodal, Sham) × 2 (Condition: Same-shape, Change-shape) mixed-design ANOVA, which revealed a trend-level interaction between stimulation and condition ($F(2,87) = 2.659$, $p = 0.076$, $\eta p^2 = 0.058$), with no significant main effects of stimulation or condition. To directly examine polarity-dependent effects, we conducted planned 2 (Group: Active vs. Sham) × 2 (Condition) ANOVAs. As shown in Fig. 7b, the comparison between the anodal and sham groups yielded a significant interaction ($F(1,58) = 4.518$, $p = 0.038$, $\eta p^2 = 0.072$). Paired $t$-tests showed that within the anodal group, RTs were significantly longer in the same-shape condition than in the change-shape condition (MD = −15.14 ms, 95% CI = [−27.37, −2.91], $t(29) = −2.531$, $p = 0.017$, Cohen's d = −0.462), indicating stronger inhibitory effects on the attended shape. In contrast, the sham group showed no significant difference between conditions (MD = 3.43 ms, 95% CI = [−9.60, 16.46], $t(29) = 0.539$, $p = 0.594$, Cohen's d = 0.098). No significant interaction was observed between cathodal and sham groups ($F(1,58) = 1.039$, $p = 0.312$, $\eta p^2 = 0.018$), and there was no significant difference in RT between the two conditions in cathodal group (MD = -4.60 ms, 95% CI = [−14.09, 4.89], $t(29) = −0.991$, $p = 0.330$, Cohen's d = −0.181). These findings align with the first tDCS experiment, suggesting that anodal tDCS enhances the inhibitory control on attended-without-memory information, providing causal evidence for the dissociation mechanism across different types of stimuli.

## Causal modulation via neuronavigated TMS supports SMG's inhibitory role

To provide more spatially precise causal evidence for the inhibitory function of the SMG, we conducted a TMS experiment using

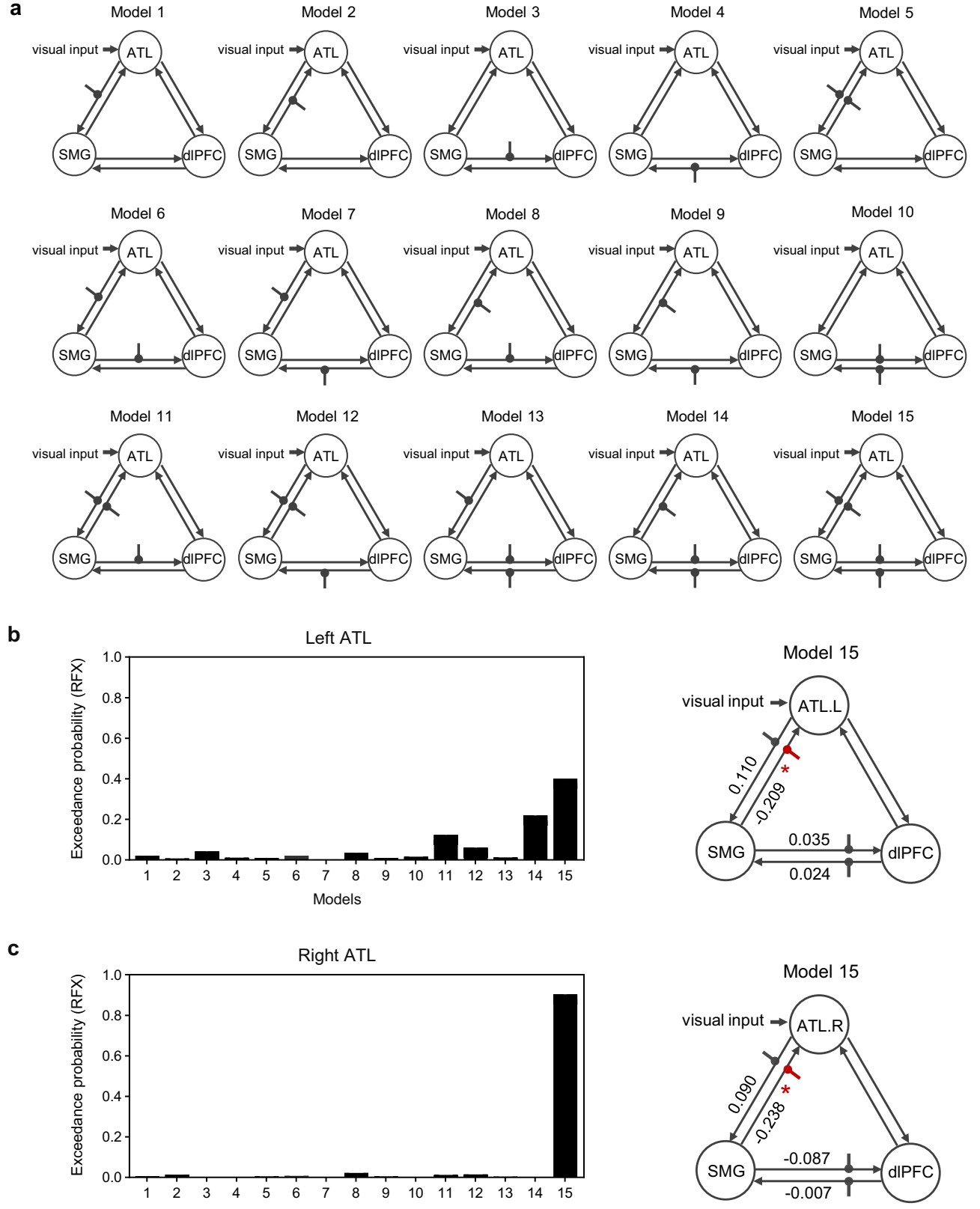

theta-burst stimulation (TBS). Continuous theta-burst stimulation (cTBS) and intermittent theta-burst stimulation (iTBS) were employed to transiently suppress or enhance cortical excitability, respectively[45]. Ninety new participants were randomly assigned to one of three groups: active cTBS, active iTBS, or sham stimulation. All participants then performed a face-based change-detection task (Fig. 6c), identical

in design to the first tDCS experiment, in which face identity served as attended-without-memory information.

A 3 (Stimulation Group: cTBS, iTBS, Sham) × 2 (Condition: Same-identity, Change-identity) mixed ANOVA revealed a significant interaction between stimulation and condition ($F(2,87) = 5.586$, $p = 0.005$, $\eta p^2 = 0.114$), a significant main effect of condition ($F(1,87) = 5.526$,

**Fig. 5 | DCM results for inhibitory circuits regulating face identity representation. a** Fifteen different models among the ATL, SMG, and dlPFC were defined for modeling the modulatory effect of the pre-surprise condition on face identity processing. The dot-headed arrows illustrate the modulatory effect of the pre-surprise condition on interregional connectivity, including the connections from the ATL to the SMG, from the SMG to the ATL, from the SMG to the dlPFC, and from the dlPFC to the SMG. **b** Left: exceedance probabilities for the 15 models among the SMG, left ATL, and dlPFC. Right: group-level mean strength of modulatory effects on interregional connectivity in the winning Model 15. The asterisk marks the only significant modulatory effect on the SMG→left ATL connection ($p = 0.017$; one-sample two-sided t test; uncorrected for multiple comparisons). **c** As in panel (**b**), but for the right ATL (ATL.R). The asterisk marks the significant modulatory effect on the SMG→right ATL connection ($p = 0.033$). *$p < 0.05$, $n = 49$. Source data are provided as a Source Data file.

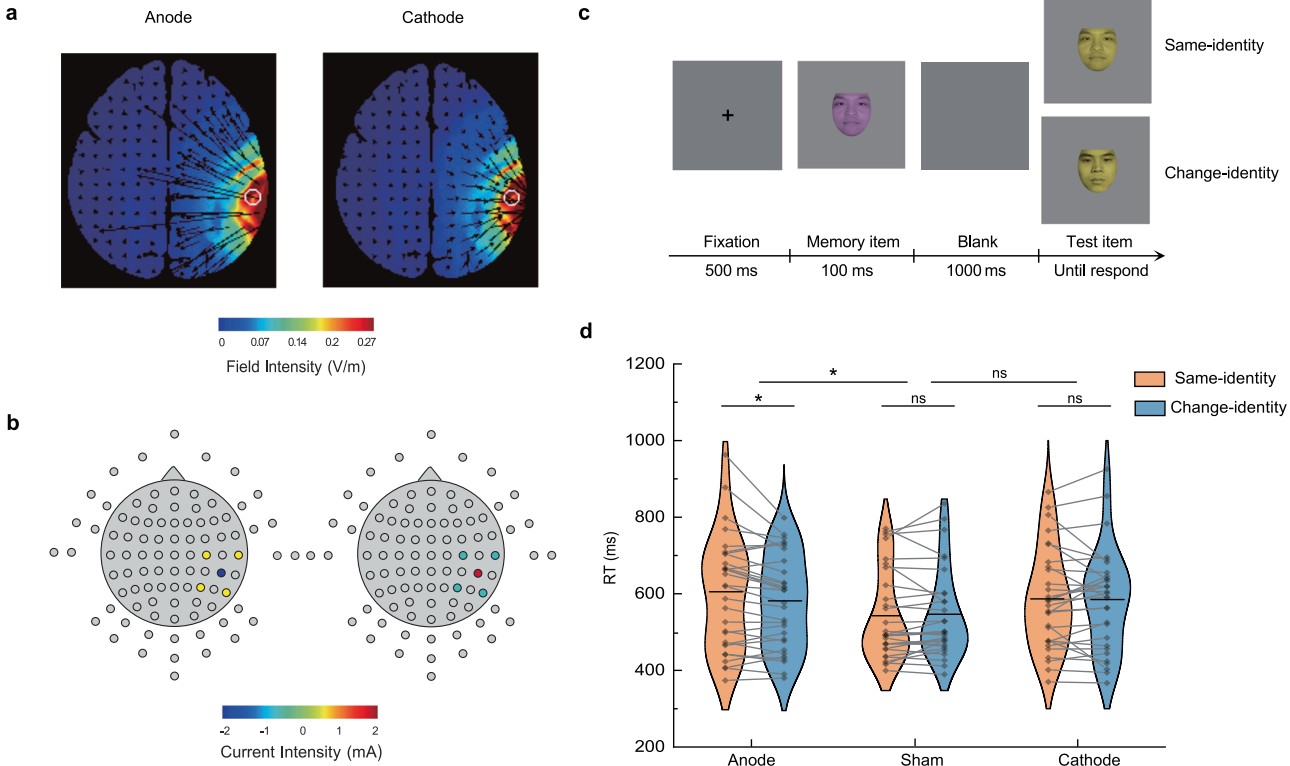

**Fig. 6 | tDCS modulation setup and results of the first tDCS experiment. a** The direction and intensity of the electrical currents were optimized for high-resolution anodal and cathodal stimulation targeting the SMG. **b** Electrode configuration, showing the central electrode at CP6, surrounded by electrodes at C4, P4, P8, and T8, along with the applied current intensity. **c** Procedure of the modified change-detection task. Participants first judged whether the memory item's identity matched a pre-specified one. If it did not, they had to memorize its color and report whether the color of the test item differed from that of the memory item. The identity of the memory item served as attended-without-memory information. **d** Change-detection task results across three post-tDCS groups. RT (reaction time): a longer RT in the same-identity condition indicates stronger inhibition of attended face identity. Violin plots show mean RT (horizontal line) and individual RTs (scatter points) linked across conditions for each participant. Planned comparison (anodal vs. sham): mixed ANOVA (Group × Condition) showed a significant interaction, $F(1,58) = 6.465$, $p = 0.014$, $\eta p^2 = 0.100$ ($n = 30$ per group). Post hoc (anodal group, two-sided paired t-test): $t(29) = -2.725$, $p = 0.011$, Cohen's d = −0.498. *$p < 0.05$; ns = not significant; uncorrected for multiple comparisons. Source data are provided as a Source Data file. Face images used under license from the Chinese Affective Face Picture System[76].

$p = 0.021$, $\eta p^2 = 0.060$), and no significant main effect of group ($F(2,87) = 0.645$, $p = 0.527$, $\eta p^2 = 0.015$). To directly examine the modulatory effects of stimulation, we compared each active stimulation group with the sham group. As shown in Fig. 8, the cTBS group showed a significant interaction with sham group ($F(1,58) = 8.211$, $p = 0.006$, $\eta p^2 = 0.124$). Specifically, in the cTBS group, RTs in the same-identity condition were significantly longer than those in the change-identity condition (MD = −23.54 ms, 95% CI = [−35.14, −11.95], $t(29) = -4.153$, $p < 0.001$, Cohen's d = −0.758), indicating enhanced inhibitory control over the residual identity representations. No such difference was observed in the sham group (MD = 0.34 ms, 95% CI = [−12.16, 12.83], $t(29) = 0.055$, $p = 0.956$, Cohen's d = 0.010), consistent with baseline inhibitory effect. In contrast, the iTBS group showed no significant interaction effect with the sham group ($F(1,58) = 0.005$, $p = 0.943$, $\eta p^2 = 0.000$), and there was no significant difference between conditions (MD = −0.26 ms, 95% CI = [−11.49, 10.97], $t(29) = -0.047$, $p = 0.963$, Cohen's d = −0.009). Taken together, these

TMS findings provide further causal evidence for the SMG's inhibitory role, with enhanced spatial precision.

## Discussion

This study provides multimodal evidence demonstrating that attentional selection and WM encoding are neurally dissociable processes, and identifies the underlying mechanisms that regulate the transition of attended neural representations into WM. Specifically, neuroimaging evidence combined with dynamic causal modeling identifies the SMG as the key region enabling this dissociation, which actively inhibits neural representations of attentional inputs unnecessary for memory, thereby preventing their entry into WM. Furthermore, findings from tDCS and TMS experiments provide causal validation of this dissociation mechanism, showing that enhancing SMG activation strengthens its inhibitory control over attended inputs and reduces their access to WM encoding.

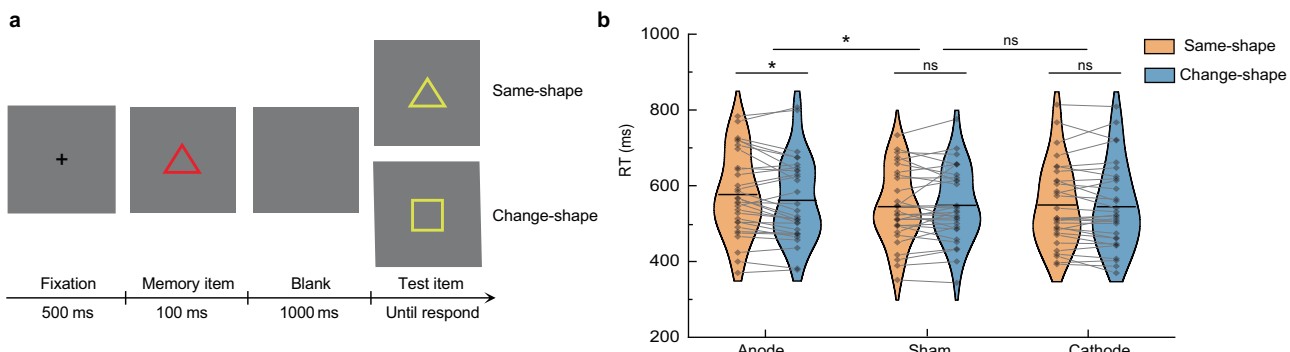

**Fig. 7 | The second tDCS experiment. a** The procedure was identical to the first experiment, but colored shapes replaced colored faces. Participants first judged whether the memory item's shape was a circle. If it was not, they memorized its color and reported whether the color of the test item had changed relative to the memory item. The shape identity of the memory item served as attended-without-memory information. **b** Ninety new participants were divided into three groups, and the results of the change-detection task after tDCS stimulation are shown for each group. Planned comparison (anodal vs. sham): mixed ANOVA (Group × Condition) showed a significant interaction, $F(1,58) = 4.518$, $p = 0.038$, $\eta p^2 = 0.072$ ($n = 30$ per group). Post hoc (anodal group, two-sided paired $t$-test): $t(29) = -2.531$, $p = 0.017$, Cohen's d $= -0.462$. *$p < 0.05$; ns = not significant; uncorrected for multiple comparisons. Source data are provided as a Source Data file.

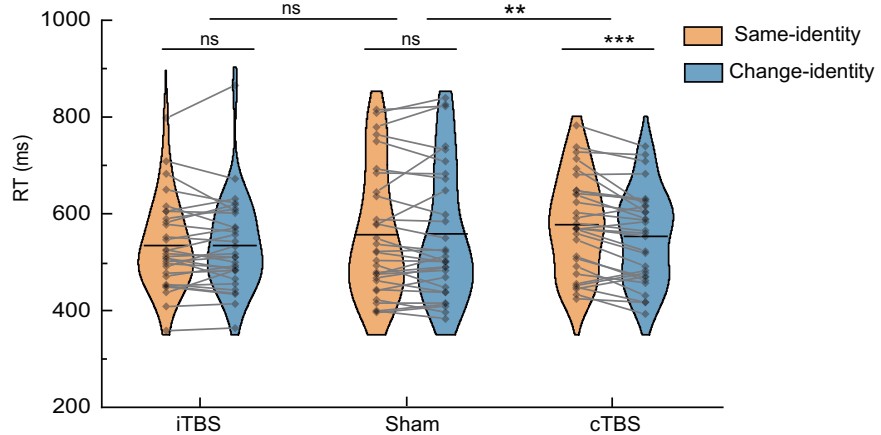

**Fig. 8 | Results of the TMS experiment.** Ninety new participants were randomly assigned to one of three groups: intermittent theta-burst stimulation (iTBS), continuous theta-burst stimulation (cTBS), or sham stimulation. The bar graph shows reaction times (RT) from the face-based change-detection task performed after stimulation in each group. Planned comparison (cTBS vs. sham): mixed ANOVA (Group × Condition) showed a significant interaction, $F(1,58) = 8.211$, $p = 0.006$, $\eta p^2 = 0.124$ ($n = 30$ per group). Post hoc (cTBS group, two-sided paired $t$-test): $t(29) = -4.153$, $p < 0.001$, Cohen's d $= -0.758$. **$p < 0.01$; ***$p < 0.001$; ns = not significant; uncorrected for multiple comparisons. Source data are provided as a Source Data file.

These findings challenge the long-standing view of the relationship between attention and WM encoding, demonstrating that they constitute two dissociable stages of information selection rather than an overlapping process. While previous studies have suggested a shared neural basis for attention and WM encoding[3–6,46], these results may stem from experimental designs that fail to disentangle these processes. Through a carefully designed experimental paradigm, we are able to directly investigate the neural dissociation between attentional selection and WM encoding. Crucially, the task design ensured that the target remained at the focus of attention, yet behavioral results showed that this attended information was not encoded into WM when participants had no expectation to report it. This observation, along with prior research on attribute amnesia[12], provides a solid foundation for establishing the neural separation between these two processes. Building on this, we identify the SMG as the central hub of the dissociation mechanism between attention and WM encoding through its active inhibition of attended inputs that are unnecessary for memory storage. Consistent with our findings, previous neurophysiological and lesion studies provide additional support for the SMG's role in inhibitory control within cognitive processes[47–51].

The dissociation mechanism redefines our understanding of how WM encodes attended inputs, revealing a dedicated gating mechanism that actively regulates encoding, rather than allowing automatic entry. The traditional view of WM gating suggests that it automatically encodes attended inputs and later actively suppresses the neural activity of outdated content, facilitating its removal from WM to optimize storage capacity[3,33,34]. In contrast, our findings reveal a distinct gating mechanism that underlies the regulation of attended neural representations as they transition into WM. Specifically, our hypothesis-driven dynamic causal modeling (DCM, Fig. 5) revealed that the SMG exerts inhibitory control over the ATL, a region involved in perceptual encoding, rather than over the dlPFC, which supports memory storage. Importantly, this inhibitory modulation effect was also observed in a data-driven, six-node DCM model. This extended model included bilateral SMG, ATL, and dlPFC regions with full connectivity and modulation (Supplementary Results and Supplementary Fig. 3), underscoring the robustness of this gating mechanism across modeling frameworks. These findings suggest that, rather than relying on storage-based removal as previously proposed, this gating mechanism selectively inhibits unnecessary perceptual inputs to

prevent their entry into WM. Supporting this interpretation, a parallel study demonstrated that attended information without memory requirements minimally impacts memory capacity in a dual-task design and lacks an online memory trace in electroencephalography[52]. Additionally, diffusion tensor imaging tractography identified anatomical pathways supporting inhibitory signal transmission from the SMG to the ATL, providing structural evidence for this gating mechanism[53,54].

Furthermore, the gating mechanism for attentional inputs into WM exhibits neural plasticity, as demonstrated by our tDCS findings. Enhancing SMG activation amplifies its inhibitory control over attended inputs that do not require memory, providing causal evidence for this mechanism. Specifically, anodal tDCS enhances activity in the stimulated region[40] and strengthens its functional connectivity with other regions[55,56], likely reinforcing the synaptic transmission of inhibitory signals from the SMG to cortical areas representing non-memorized attentional inputs. As shown in Fig. 6d, sham tDCS maintained balanced inhibitory control, keeping the neural representations of attended identities at baseline levels, consistent with previous behavioral findings[13]. In contrast, anodal tDCS heightened suppression of these attended representations, reducing them below baseline, leading to weaker memory traces for attended face identities than for unattended ones. A follow-up tDCS experiment confirmed this effect, showing a consistent amplification of inhibitory gating over shape identities (Fig. 7), demonstrating the generalizability of this mechanism from complex to simpler stimuli. Critically, this inhibitory gating effect was further corroborated by a TMS experiment, which replicated and extended the tDCS findings with higher spatial specificity (Fig. 8). These findings underscore the importance of a balanced gating mechanism, as excessive suppression may not benefit memory efficiency. Importantly, the observed functional plasticity highlights the potential for targeted interventions to regulate attentional inputs into WM, offering insights into addressing memory deficits caused by insufficient suppression of irrelevant inputs.

Additionally, the brain's gating of attended inputs that do not require WM storage is selective. According to our findings, when face reporting is not required, this gating mechanism selectively suppresses identity representations in the ATL, whereas category representations in the FFA remain unaffected. This selectivity raises a critical question: why doesn't the brain inhibit all attended representations unnecessary for memory to further reduce interference with WM? Previous research suggests that inhibition is an active suppression process that requires substantial executive control resources[57–59], with inhibitory circuits mediating synaptic transmission of neurotransmitters like GABA demanding significant energy expenditure[53,60,61]. Thus, the gating mechanism likely incurs considerable energy costs. At the same time, encoding unnecessary information into WM can create interference and compete for limited processing capacity. To optimize resource utilization, the brain appears to prioritize the inhibition of high-interference information, such as identity representations that occupy substantial capacity, while preserving lower-interference information, such as basic-level categories, which exert minimal cognitive load[15,62]. This selective inhibition reflects an adaptive trade-off between minimizing interference and conserving cognitive resources.

Several important considerations should be noted. First, while anodal tDCS demonstrated a significant enhancement effect, the expected reduction effect from cathodal tDCS was not observed, consistent with polarity-specific responses reported in prior studies[41,43,44,63–65]. This may reflect compensatory mechanisms within brain networks[64,66], or limitations associated with the offline, single-session protocol[67,68]. Future studies could employ an online approach where stimulation is delivered concurrently with behavioral testing, or adopt multiple-session protocols to amplify the modulation effect.

Second, although cTBS is generally associated with reduced cortical excitability, our TMS experiment demonstrated that cTBS over the SMG produced increased functional effects—specifically, a significant enhancement of inhibitory control. This paradoxical facilitation is consistent with prior reports showing that cTBS can sometimes improve cognitive performance in a task- and region-specific manner[69–71]. Such enhancement may be related to rebound excitability, state-dependent effects, or compensatory rebalancing within functional networks, underscoring the non-linear and context-dependent nature of stimulation effects in higher-order cortex[69,72]. Finally, our sample consisted primarily of young adult university students from Guangzhou, China, yielding a restricted age range and relatively homogeneous educational background; therefore, generalizability to other age groups, educational background, and cultural contexts may be limited. Future work should test whether SMG-mediated inhibitory gating generalizes across more diverse populations.

## Methods

### Participants

All participants were recruited from universities in Guangzhou, China, including both undergraduate and graduate students. They had normal or corrected-to-normal visual acuity, no deficits in facial recognition or color vision, and no prior experience with similar experiments. The study was approved by the Research Ethics Board of the School of Psychology at South China Normal University. All participants provided informed consent and received monetary compensation for their time. A total of 55 participants were recruited for the fMRI study. Six participants were excluded from the analysis, including three who did not complete fMRI scanning and three who exhibited excessive head motion (i.e., over 3 mm in translation or 3 degrees in rotation). The remaining 49 participants (31 females and 18 males, mean age: $20.18 \pm 2.01$ years) were included in fMRI analyses. Sex/gender information was recorded by self-report. As sex/gender was not a factor of interest in our study design, no sex/gender-based analyses were pre-specified, and the study was not powered to detect such effects.

We performed an a priori power analysis using G*Power 3.1[73] to estimate the appropriate sample size for the tDCS and TMS experiments. The expected effect size for the change-distracting effect was estimated as the effect size (dz) = 0.67 based on pilot experiments[13]. The analysis indicated that 20 participants would be sufficient to detect this effect with 80% power ($\alpha = 0.05$). Given that tDCS typically produces smaller behavioral effects[74], we conservatively increased the sample size to 30 participants per group for the tDCS experiments. The TMS experiment likewise included 30 participants per group.

For the tDCS experiments, participants were randomly assigned to the anodal, cathodal, or sham stimulation groups[69,71,75]. In the first tDCS experiment, 98 participants were initially recruited. Eight were excluded either for requesting to abort the experiment or for having a mean RT that exceeded 2.5 SD from the group mean. This left 30 participants in the anodal group (14 females and 16 males, mean age $20.53 \pm 1.61$ years), 30 in the cathodal group (12 females and 18 males, mean age $20.53 \pm 1.36$ years), and 30 in the sham group (19 females and 11 males, mean age $20.63 \pm 1.79$ years). Similarly, in the second tDCS experiment, 97 new participants were recruited. Seven were excluded using the same criteria. The remaining participants were divided into an anodal group (19 females and 11 males, mean age $20.23 \pm 1.76$ years), a cathodal group (22 females and 8 males, mean age $20.47 \pm 2.22$ years), and a sham group (19 females and 11 males, mean age $20.50 \pm 2.27$ years). In the TMS experiment, 98 new participants were initially recruited. Eight were excluded based on the same exclusion criteria used in the tDCS experiments. This yielded a final sample of 90 participants, randomly assigned to one of three stimulation groups: iTBS (14 females, 16 males; mean age $25.00 \pm 3.85$ years), cTBS

(20 females, 10 males; mean age 24.17 ± 2.68 years), and sham stimulation (22 females, 8 males; mean age 24.83 ± 2.13 years).

## Attribute amnesia task in fMRI experiment

The modified attribute amnesia task was conducted using a slow event-related design, comprising four runs (as shown in Fig. 1a). Each run consisted of 32 trials, with an inter-trial interval of was 8–12 s. As shown in Fig. 1b, each trial began with the fixation display, which consisted of a black central fixation cross and four placeholder circles positioned in the four corners of an invisible square for 1 s. Subsequently, four images including one face and three houses appeared for 1 s, which were replaced by the masks for 100 ms. A blank screen with a fixation cross followed, which presented for varying durations between 8–12 s. The response tasks differed in regular trials and surprise trials. In the regular trial, participants were instructed to report the location of the target face by pressing one of four number keys (1-4) corresponding to the four locations, then the feedback shown for 500 ms. In the surprise trial, the instruction for the face selection task was presented for 2 s to prompt participants to read it before responding, then below the instruction, four different faces appeared randomly with the numbers 1–4 until participants reported the target identity by pressing a corresponding number key. After the face selection task, participants were also asked to report the target location. Subsequently, the feedback for both tasks was displayed for 800 ms. The stimuli contained a total of four stranger-face pictures and 12 house pictures, which were processed into the same shape and grayscale[76]. The external features of faces (including hair or natural overall contours) have been removed, with the internal features of images (including eyes, nose, and mouth) serving as the primary source of information for the perceptual processing of faces[17,77]. The copyright for the Chinese Affective Face Picture System (CAFPS) used in the study has been duly purchased[76].

The experiment consisted of two conditions: a pre-surprise condition and a post-surprise condition. In both, participants performed the same face location-reporting task. In the pre-surprise condition, participants completed 60 regular trials in which they were explicitly instructed to remember only the location of the face and to report the corresponding location number. Thus, the face stimuli were fully attended but not expected to be remembered, serving as attended-without-memory information. This was followed by four surprise trials. In the first trial, participants were unexpectedly asked to report the identity of the target face, assessing whether the attended face had been encoded into working memory despite no prior expectation. The subsequent three trials served as catch trials to reinforce the expectation of identity reporting. In the post-surprise condition, participants continued the same location-reporting task, now under the assumption that face identity might be tested again. This expectation, maintained through randomly inserted surprise trials, led participants to actively store the face identity in working memory—thus, the same stimuli now functioned as attended-with-memory information.

## fMRI data acquisition

The MRI data were acquired using a Siemens 3 T Magnetom Prisma Fit scanner. Blood oxygen level-dependent (BOLD) signals were measured with an echo-planar imaging (EPI) sequence (TR/TE/flip angle = 2000 ms/30 ms / 90°, FOV = 192 × 192 mm², matrix size = 64 × 64, slice thickness = 3 mm, number of slices = 32). A high-resolution T1-weighted structural image (TR/TE/flip angle = 2530 ms/2.27 ms /7°, FOV = 256 × 256 mm², matrix size = 256 × 256, slice thickness = 1 mm, number of slices = 208) was acquired for each subject. The stimuli for the EPI scans were back-projected via a video projector (refresh rate: 60 Hz; spatial resolution: 1024 × 768) onto a translucent screen placed inside the scanner bore using MATLAB 2018b with Psychtoolbox-3[78].

The participants viewed the stimuli through a mirror located above their eyes. Each participant completed four runs of the attribute amnesia task and four runs of the face-localizer task.

## fMRI data preprocessing

The AFNI software package (version 21.2.03, http://afni.nimh.nih.gov/afni) was employed for fMRI data analysis. The fMRI preprocessing steps included: slice timing correction; head motion correction using realignment on functional volumes; co-registration of the functional image with the structural image; non-linear transformation to the Montreal Neurological Institute (MNI) template; functional volumes were resampled into 3 × 3 × 3 mm³ resolution; volumes with excessive motion were censored (marked as zeros, otherwise as ones) if the Euclidean norm of derivatives of motion parameters exceeded 0.3 mm; spatial smoothing was applied using a Gaussian filter with a 4 mm full-width half-maximum; and finally, the voxel time series were scaled to percent signal change.

## Whole-brain analyses in the attribute amnesia experiment

A whole-brain analysis was conducted using a general linear model (GLM) with the TENT function of 3dDeconvolve in AFNI[79]. The GLM analyzed the regressors of two experimental conditions (pre-surprise condition: the stimulus onset times of the pre-surprise regular trials, and the post-surprise condition: the stimulus onset times of the post-surprise regular trials) to estimate the event-related BOLD signals after stimulus onset. To control for the baseline and constrain the fitting, three additional task regressors (response onset times of pre-surprise regular trials, response onset times of post-surprise regular trials, as well as the stimuli and response onset times of all surprise trials), six motion regressors (three rotations and three translations), and one binarized censored time series (0 indicates censored time points, and 1 indicates non-censored) were included, resulting in a GLM with 12 regressors. All regressors were convolved with the canonical hemodynamic response function (HRF) to estimate the BOLD signals for each event. The TENT function, a piecewise linear spline function, was used to flexibly estimate the event-related BOLD time courses. The time courses with seven time points (from 0 to 12 s with the time step equal to TR 2 s) were modeled by TENT (0,12,7) for each event, and seven beta values corresponding to the time points were obtained. The signal value for the two conditions was defined using the average of the beta value of the peak and the preceding and following time points (4 s, 6 s, and 8 s in the time course)[80,81].

For group-level analysis, paired t-tests were conducted to compare pre- vs. post-surprise conditions. Activation maps were thresholded at $p < 0.01$ (FDR-corrected at the voxel level), with a cluster extent threshold of > 40 voxels. Significant activation clusters were defined as suprathreshold voxel clusters surrounding the peak activation voxel, with coordinates reported in MNI space using LPI orientation (Supplementary Table 1). The resulting clusters included the right SMG (68 voxels, peak at [55, −44, 37]), left DLPFC (121 voxels, peak at [−44, 25, 22]), left SMA (93 voxels, peak at [−2, 13, 58]), and left vmPFC (62 voxels, peak at [−5, 25, −20]).

## Face-localizer task in fMRI experiment

After the attribute amnesia task, a face-localizer task[26] was performed to localize the brain regions activated during face perception. The localizer task comprised four block-designed runs, each consisting of ten blocks (five face blocks and five house blocks) in randomized order. Blocks were separated by 12 s rest periods during which a fixation cross was presented. Each 12 s block comprised 24 trials, with a picture presented for 400 ms followed by a fixation cross for 100 ms. Participants were asked to perform a one-back identification task, judging whether the current picture was the same as the previous one, to maintain their focus on the stimuli.

## Localization and activation analysis of face perception regions

To localize the ATL and FFA, we employed GLM analysis to contrast the activation to face versus house stimuli in the face-localizer experiment. The GLM with the BLOCK (12,1) function of 3dDeconvolve was used to estimate BOLD signal changes from baseline for each event condition. The GLM analysis comprised nine regressors, including two task regressors (onset times of face blocks and house blocks), six motion regressors (three rotations, three translations), and one binarized censored time series (zero indicated censored time points, and one indicated non-censored). A paired t-test map at the group level was obtained for the contrast "face > house" (threshold $p < 0.01$, FDR correction at the voxel level, cluster > 40 voxels). Based on established anatomical criteria[22], the ATL was identified in the anterior part of the temporal lobe and the FFA in the mid-fusiform gyrus. ROIs were defined as suprathreshold clusters surrounding the peak voxel (in MNI space, LPI orientation). The final ROIs included: left ATL (65 voxels at [−32, −11, −38]), right ATL (93 voxels at [31, −2, −50]), left FFA (41 voxels at [−47, −56, −23]), and right FFA (110 voxels at [46, −59, −20]). These functionally defined ROIs were used in subsequent analyses.

To further investigate the neural responses to face perception in the pre-surprise and post-surprise conditions, we performed a ROI analysis[26]. We used left/right ATL and FFA separately as masks to calculate the beta values of the two conditions based on the GLM results of the attribute amnesia experiment[80]. For each ROI, the mean beta values at each TR shown in Fig. 4.

## Dynamic causal modeling analysis

ROIs were functionally defined using group-level statistical contrasts from both the attribute amnesia task and the face-localizer task. Three key regions were identified as ROIs: the SMG, representing the inhibitory control region; the left/right ATL or FFA, as core face processing regions; and the dlPFC, as the canonical region associated with working memory processing[36]. This ROI selection was guided by our central hypothesis—namely, whether the SMG suppresses perceptual information before it enters working memory, or whether such information is first encoded and then actively removed from memory storage. For each participant, BOLD signal time series were extracted by applying the group-level activation clusters as functional masks and computing the mean time-course within each ROI[82]. This approach, rather than using SPM's Volume of Interest (VOI) module to extract the principal eigenvariate time series, was chosen to ensure greater anatomical robustness and cross-individual consistency when applying functionally constrained clusters as ROIs. Details of an additional eigenvariate-based extraction using the VOI module are provided in the Supplementary Information.

A DCM analysis of the effective connectivity between ROIs under the modulatory effects of the pre-surprise condition was conducted using SPM12[27]. The analysis was performed in two stages. In the first stage, the model estimation process involved identifying the parameters that achieved an optimal balance between accuracy and complexity[37]. DCM analysis involves three sets of parameters: parameters C represent the driving input, which refers to the stimuli input into one or more regions, causing change in neural activity throughout the network; parameters A denote the baseline intrinsic connections between the regions influenced by the driving input; parameters B indicate the modulatory effects of experimental manipulations on the specified intrinsic connections. In the second stage, Bayesian model comparison was applied to identify the model that best explained the data, corresponding to the model with the minimal discrepancy between predicted and observed time series.

In our DCM analysis, as shown in Fig. 5a, the first stage involved considering parameters C, A, and B. For parameters C, we investigated the visual influence of the pre-surprise and post-surprise conditions on neural activity within the connectivity network, setting the stimuli onsets of both conditions as driving input to the left/right ATL, given

its established role in bottom-up perceptual processing. For parameters A, we used full connections, including inter-region and self-connections of regions[37,83]. For parameters B, we set the pre-surprise condition as modulatory input to assess its effect on the connectivity between SMG and the other two ROIs[31]. Specifically, based on our hypotheses, we focused on the modulatory effect of the pre-surprise condition on four directional connections: the bidirectional connection (i.e., feedforward and feedback) between SMG and ATL, as well as between SMG and dlPFC. The modulatory input could affect the connection from ATL to SMG (model 1); from SMG to ATL (model 2); from SMG to dlPFC (model 3); from dlPFC to SMG (model 4); two of the four connections (models 5–10); three of the four connections (models 11–14); and all four connections (model 15).

In the second stage, model comparison was performed by calculating the exceedance probability of each model and using the Random Effects Bayesian Model Selection[28,38]. This process evaluated the probability that a given model was more likely than any other included model from the generated data from randomly selected participants. Following the selection of the optimal model, we extracted all the modulatory effect values of the selected model from 49 individuals and analyzed the modulatory effect on each inter-regional connection using one-sample t-test.

The DCM analysis for face category processing followed a similar procedure to that for face identity processing, except that the ATL was replaced by the FFA, as shown in Supplementary Fig. 1.

In addition to the hypothesis-driven DCM analysis reported in the main text, we conducted an extended, data-driven analysis using a fully connected full-modulation model to explore condition-specific modulatory interactions among six bilateral regions of interest (ROIs): the left and right SMG, ATL, and dlPFC (Supplementary Tables 2–3 and Supplementary Fig. 3).

## HD-tDCS experiments

High-definition transcranial direct current stimulation (HD-tDCS), as an advanced, non-invasive brain stimulation technique, modulates cortical excitability in a focused and intensive manner by concentrating the direct current within a defined ring over the target areas[84]. Typically, anodal tDCS enhances cortical excitability, cathodal tDCS reduces it, and sham tDCS serves as the control[40]. In our study, we utilized HD-tDCS (Soterix Medical, New York, USA) to modulate the excitability of the SMG (coordinates as illustrated in Supplementary Table 1). The electrode montage was designed using HD-Explore Version 6.0 software with the MNI-152 template to optimize current flow to the SMG and estimate the induced electric field (Fig. 6a). For anodal tDCS, a $4 \times 1$ multielectrode anode-center cathode-surround montage was used, centered on the right SMG (CP6), with surrounding electrodes at C4, P4, P8, and T8, following the 10-10 electrode site placement[39]. Conversely, a cathode-center anode-surround montage was used for cathodal tDCS. Participants in the active group underwent 20 min of 2.0 mA stimulation with a 30 s ramp up and down). In the sham group, the electrodes were designed with half as in the anodal group and half as in the cathode group, but no stimulation was applied for 20 min except for a 30 s ramp up and a 30 s ramp down at the beginning and end of the procedure.

Following tDCS stimulation, participants performed the change-detection task and then completed the Questionnaire of Sensations Related to Transcranial Electrical Stimulation[42]. This questionnaire recorded the sensations experienced by participants during the tDCS session, such as itching, pain, and burning. In the first tDCS experiment, the questionnaire analysis revealed no significant differences in reported sensations between the Anode, Sham, and Cathode groups ($F_{2,87} = 2.364$, $p = 0.104$). Similarly, in the second experiment, no significant differences were found between the tDCS groups ($F_{2,87} = 1.214$, $p = 0.302$). In addition, participants were asked to guess the type of stimulation they had received (active vs. sham), and their responses

were approximately evenly distributed across conditions, suggesting effective blinding[85].

## Neuronavigated TMS experiment

We conducted a neuronavigated transcranial magnetic stimulation (TMS) experiment to provide spatially precise causal evidence for the role of the SMG in inhibitory control. To enable precise targeting, high-resolution structural MRI images were acquired for each participant using a 3 T Siemens Verio scanner (Germany) with a T1-weighted MPRAGE sequence (TR = 2300 ms, TE = 2.19 ms, flip angle = 9°, FOV = 256 × 256 mm², slice thickness = 1 mm, 176 slices). The stimulation target (MNI coordinates: x = 55, y = −44, z = 37) corresponded to the right SMG identified in our prior fMRI results. This location was individually localized using Visor 2.0 neuronavigation software (ANT Neuro, Netherlands) and tracked with a Polaris Spectra motion tracking system (NDI, Canada). Minor anatomical adjustments were made when necessary to account for individual variability. Magnetic stimulation was delivered via a figure-of-eight coil (outer diameter: 70 mm) connected to an NS5000 stimulator (YIRUIDE, Wuhan, China). Stimulation intensity was set to 80% of each participant's resting motor threshold (RMT), defined as the minimum intensity required to elicit a motor-evoked potential of ≥50 μV in the left first dorsal interosseous muscle in at least 5 out of 10 trials during rest.

Participants were randomly assigned to one of three stimulation groups: active cTBS, active iTBS, or sham TBS[69,71]. The cTBS protocol involved 600 pulses delivered continuously over 40 s (three pulses at 50 Hz every 200 ms). The iTBS protocol also comprised 600 pulses delivered over 200 s, consisting of 2-s trains (three pulses at 50 Hz every 200 ms), repeated every 10 s. For sham stimulation, the coil was positioned perpendicular (90°) to the scalp, preventing effective stimulation while preserving auditory and somatosensory sensations. Immediately following stimulation, participants completed the same change-detection task as used in the first tDCS experiment, designed to assess the inhibitory control over attended-without-memory face identity.

## Change-detection task

We employed the modified change-detection paradigm[13] to evaluate the impact of tDCS/TMS modulation of the SMG on the inhibition of attended-without-memory information. During the experiment, participants were seated approximately 50 cm from a 17-inch CRT monitor (60 Hz, 1024 × 768 screen resolution) and responded using a computer keyboard. The display background was medium gray (RGB: 128, 128, 128). In the first tDCS experiment, all stimuli in the memory and test displays were colored faces. There were five faces of each gender, each with five different colors: blue (91, 115, 213), green (112, 173, 71), yellow (255, 192, 0), red (192, 0, 0), and purple (112, 48, 160), all presented with 40% transparency. In the second tDCS experiment, the stimuli were colored shapes, which included circles, hexagons, triangles, squares, and stars. The colors were randomly selected from blue (0, 0, 255), green (0, 128, 0), yellow (255, 255, 0), red (255, 0, 0) and pink (255, 192, 203).

In the first tDCS experiment, participants completed a total of 120 trials, randomized into three blocks. In each trial (Fig. 6c), after a 500 ms fixation display, a memory item with a randomly selected colored face was presented in the center of the screen for 100 ms. Following a 1000 ms blank interval, a test item with a colored face was presented at the center of the screen until the participant responded or 2000 ms elapsed. Participants first judged whether the memory item's identity matched a pre-specified person. If it did, they only had to report whether the test item was also the same person (catching trials, would be excluded from the analysis). If the identity did not match, participants had to memorize the color of the face and report whether the color of the test item had changed compared to the memory item. In these non-match trials, the identity of the memory item was fully attended to but not required for reporting (i.e., attended-without-memory information). Critically, in half of the trials (same-identity condition), the identity of the test item remained the same as that of the memory item, whereas in the other half of the trials (change-identity condition), the identity of the test item changed. In the second tDCS experiment (Fig. 7a), the procedure was identical to the first, with colored shapes replacing the face stimuli. Participants first judged whether the memory item was a circle, making the shape identity the attended-without-memory information. If the item was a circle, they reported whether the test item was also a circle (catching trials). If it was not, they memorized its color and reported whether the color of the test item was different from that of the memory item.

## Analysis for change-detection task in the tDCS and TMS experiments

The identity non-match trials, where participants detected color with identity considered as attended-without-memory information, were analyzed. Catching trials, focused on identity detection, were excluded prior to analysis[13]. In line with previous research, the analysis focused on reaction times, with accuracy above 95%. Trials with incorrect responses and those with reaction times beyond ±2.5 SDs from the participant's condition-specific mean were excluded. In the first tDCS experiment, 8.21% of trials were excluded in the anodal group, 7.31% in the cathodal group, and 6.12% in the sham group. In the second tDCS experiment, 4.25% of trials were excluded in the anodal group, 5.09% in the cathodal group, and 6.03% in the sham group. In the TMS experiment, 8.39% of trials were excluded in the iTBS group, 8.11% in the cTBS group, and 8.67% in the sham group. Statistical analyses were performed using F-tests, with Greenhouse-Geisser corrections applied when sphericity violations occurred.

## Reporting summary

Further information on research design is available in the Nature Portfolio Reporting Summary linked to this article.

## Data availability

The tDCS/TMS data and derived neuroimaging data generated in this study have been deposited on the Open Science Framework (OSF) under accession DOI: 10.17605/OSF.IO/V2QN9, see ref. 86. Raw MRI data are protected and are not publicly available owing to privacy regulations; controlled access can be requested from the corresponding author (qin.pengmin@m.scnu.edu.cn). Source data are provided with this paper.

## Code availability

Custom code on OSF: https://doi.org/10.17605/OSF.IO/V2QN9. Other analyses used standard packages per official tutorials; no additional custom code.

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

## Acknowledgements

This work was supported by grants from Science and Technology Innovation 2030-"Brain Science and Brain-like Research" Major Project (No.2022ZD0210800 for H.C.), Emerging Enhancement Technology under Perspective of Humanistic Philosophy, supported by National Office for Philosophy and Social Science (No. 20&ZD045 for H.C.), National Natural Science Foundation of China (No.31971032, No. 32371098 for P.Q., No.32171046 for H.C., No.32200844 for Y.F.), Fundamental Research Funds for the Central Universities (226-2024-00207, 226-2024-00118), Guangdong Basic and Applied Basic Research Foundation (2024A1515011429 for P.Q.), Research Center for Brain Cognition

and Human Development, Guangdong, China (No. 2024B0303390003 for P.Q.), and Open Research Fund of the Zhejiang Key Laboratory of Precision Psychiatry (No. 2025B5 for Y. L.).

## Author contributions

Y.L. was responsible for conceptualization, data acquisition, formal analysis, drafting the manuscript, and reviewing and editing. Y.F. and E.T. contributed to conceptualization, validation, drafting the manuscript, and reviewing and editing. H.W., J.Han., M.X., and Y.Z. were responsible for data analysis, drafting the manuscript, and reviewing its content. B.P. and J.Huang. contributed to data acquisition. H.L. provided methodology guidance, supervision for analysis, and manuscript review. H.C. and P.Q. were responsible for conceptualization, funding acquisition, methodology, supervision, and manuscript review and editing.

## Competing interests

The authors declare no competing interests.
