## [Transparent Peer Review file · Nature Communications]

Neural Dissociation of Attention and Working Memory through Inhibitory Control

Corresponding Author: Professor Hui Chen

Version 0:

Reviewer comments:

Reviewer #1

(Remarks to the Author)

This study aimed to provide neural evidence that attention and working memory (WM) encoding are dissociable processes, identifying the supramarginal gyrus (SMG) as a key region supporting this dissociation. Using fMRI, dynamic causal modeling (DCM), and transcranial direct current stimulation (tDCS), the authors revealed a neural mechanism by which the SMG inhibits attentional representations from being integrated into WM. A major strength of the study is its multimodal approach, combining neuroimaging and neuromodulation to establish both correlational and causal evidence. However, limitations include the use of an oversimplified model, reliance on a relatively dated DCM technique, lack of detailed reporting on fMRI analysis methods, and a demographically homogeneous sample with limited age variability.

Major points:

1. Results: This section should be more concise. While I understand that this journal places the Results section before the Methods, a substantial amount of content currently in the Results section more appropriately belongs in the Methods. These methodological details should be moved accordingly, with only a simplified summary retained in the Results to provide necessary context. Similarly, some parts of the Results are more interpretive in nature and would be better suited for the Discussion section.
2. In the subsection “Stronger activation of the SMG in the pre-surprise condition” under Results, the authors report activations primarily in the dlPFC, vmPFC, SMG, and SMA. However, given that both working memory and attention tasks typically engage widespread neural networks, it would be important to also report other significantly activated brain regions beyond the a priori regions of interest. This would provide a more comprehensive understanding of the whole-brain activation patterns and ensure transparency in the results.
3. Regions of Interest (ROIs): The manuscript lacks a clear description of how the ROIs were defined for both the ROI-based analysis (e.g., Figure 3) and the DCM analysis. It is unclear whether these ROIs were selected a priori, what their shape and size were, and how they were constructed. Information about ROI locations only becomes available upon consulting the Supplementary Information (SI). A detailed description of ROI definition should be provided, as this is essential for transparency and replicability of the findings.
4. Laterality of SMG in Activation and DCM Analyses: It is unclear from the main text whether the SMG region analyzed in the brain activation and DCM analyses refers to the right SMG. While this can be inferred from the tDCS methods and Supplementary Information, the main text should clearly specify this. Additionally, it would strengthen the manuscript if the authors provided a rationale for focusing on the right SMG over the left, particularly given that both hemispheres may contribute to attention and working memory processes.
5. Participant Demographics: In the Participants section, the manuscript states that “The remaining 49 participants (31 females and 18 males, mean age: 20.18 ± 2.01 years) were included in fMRI analyses,” but it does not specify where the participants were recruited from. Additionally, it is unclear whether their educational background or cognitive abilities (e.g., IQ) were assessed. As the sample appears demographically homogeneous and has limited age variability, reporting these additional details would enhance the generalizability and interpretability of the findings. It would also be helpful if the authors can describe the criteria for matching the three groups: i.e., anodal group, cathodal group and sham group.
6. Whole-Brain Analyses and Software Consistency: In the Methods section, the authors describe using AFNI for whole-brain analyses in the attribute amnesia experiment. While using AFNI is entirely valid, the manuscript would benefit from a more consistent analytical workflow. Given that the study also conducted DCM analyses—typically implemented in SPM—it would be methodologically smoother and more efficient to perform whole-brain activation analyses in SPM instead of AFNI. This would facilitate the extraction of ROI time series for subsequent DCM analysis.

7. Time Series Extraction for DCM: In the Dynamic Causal Modeling analysis subsection, the authors state, "The mean time series for each ROI was extracted from our preprocessed functional data for the subsequent analyses." Typically, the standard practice in DCM is to extract the principal eigenvariate from each ROI, which captures the dominant temporal component and is more robust to noise. Although the mean time series may yield similar results, the implications of using it in place of the principal eigenvariate are unclear. This issue further underscores the concern raised in comment #6.
8. The current three-node DCM model appears overly simplistic. Rather than modeling the bilateral networks in a unified manner—which better reflects how the brain operates—the authors analyzed two separate three-node models for each hemisphere. More concerning, it seems that the right SMG was used in the model intended for the left hemisphere, which raises further questions. Why wasn't the left SMG used in this case? And why not construct a more comprehensive bilateral model with a greater number of nodes to more accurately capture inter-hemispheric connectivities?
9. Limitations of Model Specification. For each hemisphere, the authors specified 15 DCM models. However, even with only three nodes, these 15 models do not exhaust all plausible configurations. For instance, the driving input could target nodes other than the ATL, and modulatory effects could exist on the connections between the ATL and dlPFC. These modeling choices likely reflect limitations inherent to Bayesian Model Selection (BMS), which requires a pre-defined and limited model space. While BMS is feasible for small models, it becomes impractical for larger networks due to the combinatorial explosion of possible configurations. A more suitable and scalable alternative would be the use of DCM Parametric Empirical Bayes (PEB), which allows efficient exploration of model space by pruning non-contributory parameters in a data-driven manner. Unfortunately, PEB was not implemented in this study.
10. Justification for Statistical Approach in DCM Analysis. In the "Dynamic causal modeling analysis" subsection, the authors state: "we extracted all the modulatory effect values of the selected model from 49 individuals and analyzed the modulatory effect on each inter-regional connection using one-sample t-test." However, it is unclear why Bayesian Model Averaging (BMA) was not used. BMA is typically preferred over relying on a single winning model, as it accounts for model uncertainty and provides more robust parameter estimates across plausible models. The rationale for omitting BMA should be clarified.

minor points

11. Caption of Table 1 can be more detailed: it would be helpful to remind the readers that there are no surprise trials in Run 1.
12. Software Description for DCM: The statement, "A DCM analysis of the effective connectivity between ROIs under the modulatory effects of the pre-surprise condition was conducted using SPM12 toolbox," could be more accurately phrased as, "using the SPM software," since multiple toolboxes are associated with SPM.
13. The term "exceedance probability" should be briefly explained when it first appears in the manuscript.

Reviewer #2

(Remarks to the Author)

In three experiments the authors aimed to demonstrate that attention and working memory are dissociative cognitive processes which are modulated by circuitry involving inhibitory attentional control by the supramarginal gyrus (SMG). In the first study involving task related fMRI they used an attribute amnesia task to identify brain regions involved in the regulating the transition of stimuli into working memory. This study showed that increased activation of the SMG was associated with inhibiting stimuli from entering working memory. This study also showed that there was a time course difference in BOLD signal activation in the bilateral anterior temporal lobe for face perception depending on whether attended to with or without memory. The following two experiments then used two different versions of a different cognitive task (change detection task) together with high-definition transcranial direct current stimulation (HD-tDCS) to show that stimulation of the SMG could reduce performance (slow reaction time) in the same identity/shape condition compared to the change identity/shape condition. This effect was replicated for anodal tDCS in both experiments. The authors interpreted this finding as direct causal evidence of a role of the SMG in attentional inhibition on the attended face/shape. Strengths of the studies presented included the task-related fMRI experiment showing a role of the SMG in uncoupling of attention and memory and the dynamic causal modelling showing the activity in the functional network involved. A potential weakness were the HD-tDCS experiments and the assertion that these provide direct causal evidence that modulation of the SMG affected attentional inhibitory processes. The main weakness I see is the assertion from the behavioural results (reaction times) on the respective tasks supports a role of the SMG in these processes. Although the behavioural effect was replicated (which is a further study strength) both the task and small sized effect seen with the anodal conditions compared to sham are not convincing in supporting a specific role of the SMG in attention inhibition. An alternative task paradigm with a clearer behavioural effect (as seen with the attribute amnesia task with task-related fMRI) would have provided more convincing evidence. Also, a more focal form of brain stimulation (e.g., rTMS) may have been a better choice for demonstrating a causal role of this region in the effect. HD-tDCS involves stimulation (both excitatory and inhibitory) over a reasonably large cortical area which involves the SMG and several other neighbouring regions, so any specific behavioural effect (slowing of RT) in a particular condition could also potentially be attributed to modulation of other brain regions/networks other than the SMG.

I have the following queries/comments to assist with further improving the paper.

1. Results. Table 1. This data would be better presented as bar graph showing SD. Why wasn't a RMANOVA conducted to examine these behavioural results?
2. Results. Figure 2. Instead of using blue, the figures would be clearer showing the increased activation in the memory regions as red.
3. Methods. tDCS experiments 1 and 2. Please clarify whether these studies were randomised.
4. Methods. tDCS experiments. Was a power analysis conducted for these experiments? tDCS behavioural effects are very small (Dedoncker et al., 2016, Brain Stimul)

5. Results. tDCS experiments. Please provide the results from the full model ANOVA, including specific main effects and interaction effects.

6. Results. tDCS experiments. Note that reporting the instances of side effects for the different stimulation conditions is not the same as directly testing adequacy of participant blinding, e.g., asking participants whether they thought that they received active or sham tDCS. If blinding was not assessed, then this should be stated as a limitation.

Reviewer #3

(Remarks to the Author)

This is a timely study on the interaction of attentive selection and working memory, the methods are sound and the paper generally well written. I do, however, have one concern that relates to the way the authors present their findings where they stress the uniqueness of their results. First, other than stated in the first sentence of the abstracts, although everyone in the field would agree that both processes are closely related, no-one would claim that they are inseparable. Likewise, their study is not the first to provide evidence that the two processes are dissociable in terms of brain physiology (e.g. McNab & Klingberg, 2008). Hence, I recommend the authors to formulate their statements with more care.

Version 1:

Reviewer comments:

Reviewer #1

(Remarks to the Author)

The authors have addressed most of my comments. I have one additional suggestion: While the revisions to the Participants section are helpful, the authors should also acknowledge the demographic characteristics of their sample as a limitation. In particular, the restricted age range and educational homogeneity (i.e., university students in Guangzhou, China) may limit the generalizability of the findings to broader and more diverse populations. Explicitly noting this in the Discussion as a study limitation would further strengthen the manuscript's transparency and contextualization.

Reviewer #2

(Remarks to the Author)

The addition of the new rTMS experiment significantly strengthens the authors' conclusions. My other concerns and queries have been adequately addressed.

REVIEWER COMMENTS

Reviewer #1 (Remarks to the Author):

This study aimed to provide neural evidence that attention and working memory (WM) encoding are dissociable processes, identifying the supramarginal gyrus (SMG) as a key region supporting this dissociation. Using fMRI, dynamic causal modeling (DCM), and transcranial direct current stimulation (tDCS), the authors revealed a neural mechanism by which the SMG inhibits attentional representations from being integrated into WM. A major strength of the study is its multimodal approach, combining neuroimaging and neuromodulation to establish both correlational and causal evidence. However, limitations include the use of an oversimplified model, reliance on a relatively dated DCM technique, lack of detailed reporting on fMRI analysis methods, and a demographically homogeneous sample with limited age variability.

Major points:

1. Results: This section should be more concise. While I understand that this journal places the Results section before the Methods, a substantial amount of content currently in the Results section more appropriately belongs in the Methods. These methodological details should be moved accordingly, with only a simplified summary retained in the Results to provide necessary context. Similarly, some parts of the Results are more interpretive in nature and would be better suited for the Discussion section.

Response:

We sincerely thank the reviewer for this suggestion. In response, we have substantially revised the Results section to improve clarity, focus, and organization, ensuring a clearer separation between methodological descriptions, statistical outcomes, and theoretical interpretations. Specifically:

1) MRI Task Design Details: We moved the detailed three-paragraph description of the fMRI task design—including trial structure, stimulus types, and the surprise-test manipulation—from the Results to the Methods section (p. 28). In the Results, this content has been condensed into a brief summary paragraph (p. 5), retaining only essential contextual information. Key task details have also been incorporated into the caption of Figure 1 (p. 6) to support reader comprehension without interrupting the flow of results.

2) ROI Localizer: The description of the face perception localizer task has been streamlined to two concise sentences in the Results (p. 10), preserving only the purpose and criteria used for defining ROIs. Full procedural details have been relocated to the Methods (p.30).

3) tDCS Design Summary: In line with the reviewer's suggestion, we have streamlined the tDCS design descriptions in the Results section. Detailed information regarding stimulation parameters and electrode placement has been moved to the Methods section (p. 33). The Results section (p. 15) now contains a concise summary of the tDCS experiments, focusing solely on the stimulation groups, basic design, and task objectives.

4) Interpretive Content: In response to the reviewer’s suggestion that “Results are more interpretive in nature and would be better suited for the Discussion section,” we carefully reviewed the manuscript and revised accordingly. Specifically, theoretical interpretations of the DCM findings—particularly those addressing the dissociation between attention and WM encoding—have been removed from the Results section and are now fully presented in the Discussion (p. 23). Those interpretations were already included in the original Discussion and have been refined to enhance clarity and coherence.

We believe these revisions significantly improve the manuscript’s structure and readability and better align with the journal’s standards for scientific reporting.

2. In the subsection “Stronger activation of the SMG in the pre-surprise condition” under Results, the authors report activations primarily in the dlPFC, vmPFC, SMG, and SMA. However, given that both working memory and attention tasks typically engage widespread neural networks, it would be important to also report other significantly activated brain regions beyond the a priori regions of interest. This would provide a more comprehensive understanding of the whole-brain activation patterns and ensure transparency in the results.

Response:

We thank the reviewer for this suggestion. The activations reported in the subsection “Stronger activation of the SMG in the pre-surprise condition”—specifically in the SMG, dlPFC, SMA, and vmPFC—represent all the significant clusters identified from a whole-brain contrast between the pre- and post-surprise conditions. This analysis employed a stringent statistical threshold (voxel-wise FDR-corrected $p < 0.01$) with a minimum cluster extent of > 40 voxels. These regions reflect the full outcome of a data-driven group-level analysis and are not restricted to a priori ROIs.

To provide a more comprehensive view of whole-brain activation, we used a more lenient threshold (uncorrected $p < 0.005$; cluster extent > 40 voxels). This revealed supplementary clusters that did not survive multiple-comparison correction, primarily involving contralateral homologues of the previously reported regions (e.g., left SMG, right dlPFC, right SMA, and right vmPFC).

We have now included the resulting activation map in **Supplementary Fig. 2** and clearly described its exploratory nature in the figure caption.

Supplementary Fig. 2. Whole-brain activation results for the attribute amnesia task using a more lenient statistical threshold. Broader activation patterns are visualized using a group-level GLM analysis of the contrast “pre-surprise $>$ post-surprise” at a more lenient threshold ($p < 0.005$, uncorrected; cluster extent > 40 voxels).

Warm colors (red and yellow) indicate greater activation in the pre-surprise condition, while cool colors (blue and green) indicate greater activation in the post-surprise condition. Notable clusters were observed in bilateral SMG (supramarginal gyrus), dlPFC (dorsolateral prefrontal cortex), SMA (supplementary motor area), and vmPFC (ventromedial prefrontal cortex).

3. Regions of Interest (ROIs): The manuscript lacks a clear description of how the ROIs were defined for both the ROI-based analysis (e.g., Figure 3) and the DCM analysis. It is unclear whether these ROIs were selected a priori, what their shape and size were, and how they were constructed. Information about ROI locations only becomes available upon consulting the Supplementary Information (SI). A detailed description of ROI definition should be provided, as this is essential for transparency and replicability of the findings.

Response:

We thank the reviewer for this comment. In response, we have added detailed descriptions of ROI definition procedures to both the Results and Methods sections of the main text to enhance the transparency and replicability.

First, we clarified ROI construction in the Results (whole-brain GLM analyses, Fig. 3): *“A general linear model analysis was conducted to compare BOLD responses between the pre- and post-surprise conditions. A paired t-test was applied using a significance threshold of $p < 0.01$ (FDR-corrected at the voxel level) and a minimum cluster size of > 40 voxels. The resulting clusters included the right supramarginal gyrus (SMG; 68 voxels, peak at [55, -44, 37]), left dorsolateral prefrontal cortex (dlPFC; 121 voxels, peak at [-44, 25, 22]), left supplementary motor area (SMA; 93 voxels, peak at [-2, 13, 58]), and left ventromedial prefrontal cortex (vmPFC; 62 voxels, peak at [-5, 25, -20]).”* (p. 9)

Second, for the ROI-based analyses (Fig. 4), we revised the Results to include: *“To localize face-selective ROIs, participants completed a face-localizer task during the fMRI session. Based on combined anatomical and functional criteria, we successfully identified bilateral ATL and bilateral FFA. ROIs were defined as suprathreshold clusters centered on peak voxels in MNI space (LPI orientation), using a threshold of $p < 0.01$ (FDR-corrected at the voxel level). Specifically, the ROIs included the left ATL (65 voxels at [-32, -11, -38]), right ATL (93 voxels at [31, -2, -50]), left FFA (41 voxels at [-47, -56, -23]), and right FFA (110 voxels at [46, -59, -20]).”* (pp. 10–11)

Third, for the DCM analysis (Fig. 5), we added: *“The DCM analysis focused on connectivity changes among three key regions identified in prior analyses (Figs. 3 and 4): the inhibition region (SMG), the face perception region (ATL or FFA), and the WM storage region (dlPFC). All ROIs were defined based on suprathreshold activation clusters from whole-brain group-level contrast maps, using a consistent statistical threshold (voxel-wise FDR-corrected $p < 0.01$).”* (p. 12)

Finally, we further expanded the Methods section (pp. 29–31) to provide technical details regarding the statistical thresholds, coordinate space, and shape/extent of all ROIs.

4. Laterality of SMG in Activation and DCM Analyses: It is unclear from the main text

whether the SMG region analyzed in the brain activation and DCM analyses refers to the right SMG. While this can be inferred from the tDCS methods and Supplementary Information, the main text should clearly specify this. Additionally, it would strengthen the manuscript if the authors provided a rationale for focusing on the right SMG over the left, particularly given that both hemispheres may contribute to attention and working memory processes.

Response:

We thank the reviewer for this suggestion. In response, we have now clearly specified that the SMG analyzed in both the brain activation and DCM analyses refers specifically to the right SMG.

To enhance clarity, we revised the Results section (“Stronger activation of the SMG in the pre-surprise condition”, pp. 8–10) as follows:

“We conducted a whole-brain paired t-test comparing the pre-surprise and post-surprise conditions, with statistical maps thresholded at $p < 0.01$ (FDR-corrected at the voxel level) and a cluster extent threshold of >40 voxels. As shown in Fig. 3, the right supramarginal gyrus (SMG) exhibited significantly greater activation in the pre-surprise condition, suggesting its role in inhibiting the encoding of target faces that are irrelevant to memory.”

We also updated the Figure 3 legend to include peak coordinate information (p. 9):

“Significant clusters included the right SMG (68 voxels, peak at [55, -44, 37]).”

Additionally, we clarified the rationale for lateralization in the following paragraph: (pp.9-10):

“Notably, only the right SMG, which may be associated with inhibitory function, survived the FDR-corrected threshold. To explore potential bilateral involvement, we conducted an exploratory analysis using a more lenient threshold ($p < 0.005$, uncorrected), which revealed weaker activation in the left SMG (see Supplementary Fig. 2), although this did not survive statistical correction. As the current results align with prior evidence²³ indicating a right-hemispheric dominance of parietal regions in executive control and top-down inhibition, we focused all subsequent analyses in the main text (e.g., dynamic causal modeling and neuromodulation) on the right SMG, hereafter referred to simply as SMG.”

These revisions ensure consistency in the interpretation of lateralized SMG activity across all components of the study and provide a theoretically grounded justification for our analytical focus.

5. Participant Demographics: In the Participants section, the manuscript states that “The remaining 49 participants (31 females and 18 males, mean age: 20.18 ± 2.01 years) were included in fMRI analyses,” but it does not specify where the participants were recruited from. Additionally, it is unclear whether their educational background or cognitive abilities (e.g., IQ) were assessed. As the sample appears demographically homogeneous and has

limited age variability, reporting these additional details would enhance the generalizability and interpretability of the findings. It would also be helpful if the authors can describe the criteria for matching the three groups: i.e., anodal group, cathodal group and sham group.

Response:

We thank the reviewer for this comment. In response, we have revised the “Participants” subsection of the Methods to provide additional demographic and recruitment details, thereby enhancing the interpretability and generalizability of our findings.

All participants were recruited from universities in Guangzhou, China, and consisted of both undergraduate and graduate students. Given the homogeneity of our sample—university students with comparable educational levels—we did not administer formal IQ or standardized cognitive assessments. This approach is consistent with previous studies investigating attention and working memory in similar populations (e.g., Zanto et al., *Nature Neuroscience*, 2011; Fu et al., *Science Advances*, 2021), where university samples were considered to possess typical cognitive functioning sufficient for task engagement.

For the tDCS experiments, participants were randomly assigned to one of three stimulation groups: anodal, cathodal, or sham. Randomization was employed to ensure comparability in demographic and cognitive characteristics across groups and to minimize potential confounds due to individual differences.

We have incorporated these details into the revised manuscript (Methods – Participants section, p. 26), and we are grateful to the reviewer for prompting this clarification, which helped strengthen the methodological transparency of our study.

6. Whole-Brain Analyses and Software Consistency: In the Methods section, the authors describe using AFNI for whole-brain analyses in the attribute amnesia experiment. While using AFNI is entirely valid, the manuscript would benefit from a more consistent analytical workflow. Given that the study also conducted DCM analyses—typically implemented in SPM—it would be methodologically smoother and more efficient to perform whole-brain activation analyses in SPM instead of AFNI. This would facilitate the extraction of ROI time series for subsequent DCM analysis.

Response:

We thank the reviewer for this suggestion. Our choice to conduct the whole-brain analyses in AFNI was motivated by the specific methodological advantages it offers for slow event-related designs.

In particular, AFNI’s 3dDeconvolve with TENT basis functions allowed us to flexibly model the full time course of the BOLD response without assuming a fixed shape. This approach was critical for accurately capturing event-related temporal dynamics in regions such as the FFA and ATL. While SPM also supports Finite Impulse Response (FIR) modeling, AFNI provides additional flexibility and visualization control, particularly under conditions

involving jittered trial timing and variable interstimulus intervals. Furthermore, to maintain spatial consistency, all steps of the preprocessing pipeline—including normalization, smoothing, ROI definition and analysis, as well as neuromodulation targeting—were conducted within the AFNI environment. This approach minimized potential discrepancies arising from cross-platform processing.

To directly address the reviewer’s concern, we conducted a validation analysis in SPM to assess whether our key activation results—particularly in the right SMG—would replicate using a SPM pipeline. In this analysis, we performed a whole-brain GLM using the canonical HRF and a paired t-test (pre-surprise > post-surprise) with the same statistical thresholds as in the AFNI analysis (voxel-wise FDR-corrected $p < 0.01$, cluster extent > 40 voxels).

As shown in the figure below, the SPM results revealed significant activation in the right SMG (peak at [68, -40, 38], cluster size = 88 voxels), closely matching the AFNI-derived activation (peak at [55, -44, 37], cluster size = 68 voxels). Both results are reported in MNI space using LPI orientation. This alignment between AFNI and SPM results provides converging evidence for the reliability of the SMG activation.

Figure. SMG activation results using SPM

Given the consistency across platforms and the specific advantages of AFNI for our experimental design, we retained AFNI-derived activation coordinates throughout the pipeline to ensure internal consistency. We thank the reviewer for prompting this validation step, which provides additional support for the robustness and methodological soundness of our results.

7. Time Series Extraction for DCM: In the Dynamic Causal Modeling analysis subsection, the authors state, “The mean time series for each ROI was extracted from our preprocessed functional data for the subsequent analyses.” Typically, the standard practice in DCM is to extract the principal eigenvariate from each ROI, which captures the dominant temporal component and is more robust to noise. Although the mean time series may yield similar results, the implications of using it in place of the principal eigenvariate are unclear. This issue further underscores the concern raised in comment #6.

Response:

We thank the reviewer for this comment. In our main analysis, the mean BOLD time series from each functionally defined ROI was extracted for DCM. This choice was made to ensure consistency across participants and comparability of signals: all ROIs were

defined from group-level activation clusters identified through AFNI preprocessing and whole-brain analyses, and the mean signal ensured stable, comparable time series across all participants. Similar approaches have been used in previous DCM studies (e.g., Bajaj et al., 2016).

To address the reviewer's concern, we conducted a supplementary eigenvariate-based extraction using SPM's Volume of Interest (VOI) module, following established recommendations (Zeidman et al., 2019). Spherical search volumes (outer radius = 8 mm, inner radius = 6 mm) were centered on group-level peak coordinates and thresholded at $p < 0.05$ (uncorrected), with subject-specific local maxima used to define VOIs. Valid eigenvariates across all ROIs were obtained for 22 of 49 participants. Most exclusions (21 participants) resulted from unsuccessful extraction in the left or right ATLs, which lie near the ear canal and are susceptible to magnetic field inhomogeneities (Collin et al., 2014; Rajimehr et al., 2009). Several fMRI studies have reported similar difficulties in obtaining reliable activation of ATLs at the individual level—for example, Pinsk et al. (2009) and Rajimehr et al. (2009) observed face-selective responses in only about half of their participants, whereas Tsao et al. (2008) reported such responses in less than one-third of the sample. Previous DCM studies also supported such challenges in extracting sufficient signal from certain ROIs using the standard VOI procedure (Jamieson et al., 2023; Zeidman et al., 2019).

We have clarified the rationale and implications of using the mean time series approach instead of the principal eigenvariate in the revised Methods (DCM Analysis subsection) and provided the eigenvariate-based extraction procedure and results in the Supplementary Information.

8. The current three-node DCM model appears overly simplistic. Rather than modeling the bilateral networks in a unified manner—which better reflects how the brain operates—the authors analyzed two separate three-node models for each hemisphere. More concerningly, it seems that the right SMG was used in the model intended for the left hemisphere, which raises further questions. Why wasn't the left SMG used in this case? And why not construct a more comprehensive bilateral model with a greater number of nodes to more accurately capture inter-hemispheric connectivities?

Response:

We sincerely thank the reviewer for this comment. Our modeling strategy was guided by a **hypothesis-driven** approach focused on how the brain suppresses attended representations that are unnecessary for memory (Fu et al., Trends in Cognitive Sciences, 2023). Specifically, our central hypothesis was that the SMG exerts top-down inhibitory control either over perceptual processing in the ATL or over memory storage in the dlPFC. Accordingly, we constructed three-node DCMs including the SMG, ATL, and dlPFC.

The right SMG was used because, in the group-level contrast of the attribute amnesia task (pre-surprise > post-surprise), only the right SMG survived the corrected statistical threshold (voxel-wise $p < 0.01$, FDR-corrected; cluster extent > 40 voxels). No significant

activation was observed in the left SMG at this threshold. Thus, the right SMG—a region consistently implicated in inhibitory control—was selected as the source region in our main DCM model. This selection is also consistent with prior literature highlighting right-lateralized parietal contributions to executive control and top-down inhibition (e.g., Benedek et al., 2014).

We chose to construct two separate three-node models rather than a unified six-node bilateral model for both theoretical and statistical reasons. (1) Empirical grounding: the unilateral ROIs used in the main model were derived from statistically robust clusters that survived correction. (2) Model reliability: larger DCMs with more nodes often suffer from reduced identifiability and unstable parameter estimation, especially with limited trial numbers. This tradeoff between model complexity and reliability is well documented (Friston et al., 2003; Zeidman et al., 2019). Using a restricted model space is therefore standard practice in hypothesis-driven DCM studies (Rigoux et al., 2014; Huang et al., 2023; Heimhofer et al., 2024).

Given that comments 9 and 10 raised related concerns, we additionally conducted an extended **data-driven** analysis using Parametric Empirical Bayes (PEB) with Bayesian Model Averaging (BMA). This extended model included a fully connected six-node network comprising bilateral ATL, SMG, and dlPFC, with all bidirectional intrinsic and modulatory connections. Contralateral homologues, such as the left SMG (peak coordinates [-68 -35 34]), were defined using a more lenient threshold ($p < 0.005$, uncorrected; Supplementary Fig. 2; Supplementary Table 2). Time series were extracted using the standard VOI procedure in SPM (Zeidman et al., 2019). Valid eigenvariates across all six ROIs were obtained for 22 participants, and the PEB+BMA analysis was conducted on this subset.

Supplementary Fig. 2. Whole-brain activation results for the attribute amnesia task using a more lenient statistical threshold.

As shown in Supplementary Fig. 3 and Supplementary Table 3, the strongest evidence for negative modulation (posterior probability > 95%; posterior expectation = -0.699) was observed from the right SMG to the left ATL. In contrast, no significant modulatory effects were found from the SMG to the dlPFC. These results support our hypothesis that the SMG selectively inhibits perceptual encoding (ATL) rather than memory storage (dlPFC), consistent with the findings of our original hypothesis-driven model.

Supplementary Fig. 3. Full-modulation model results from the extended DCM analysis. Six bilateral ROIs were included: left and right SMG, ATL, and dlPFC. Each region was bidirectionally connected to the others (double-headed arrows) and included a self-connection (not shown for clarity). Visual inputs from both pre- and post-surprise conditions entered the bilateral ATL, and the pre-surprise condition modulated all inter-regional and self-connections. Significant modulatory effects (posterior probability > 95%) are indicated by red arrows. Notably, the analysis revealed strong inhibitory modulation from the right SMG to the left ATL (modulatory effect = -0.699), suggesting enhanced top-down suppression of perceptual representations in ATL. No significant modulatory effects were observed from SMG to dlPFC, reinforcing the specificity of SMG–ATL interactions. Other significant connections (e.g., between bilateral homologues) were observed but are not central to our theoretical model.

Supplementary Table 3. Parameter estimates from the full-modulation model.

Connection	Modulatory effect	Posterior probability
RDLPFC → RDLPFC	-0.307	0.518
RDLPFC → RSMG	0.198	0.663
RDLPFC → RATL	0.728	1.000
RDLPFC → LDLPFC	0.402	1.000
RDLPFC → LSMG	0.000	0.000
RDLPFC → LATL	0.547	1.000
RSMG → RDLPFC	0.000	0.000
RSMG → RSMG	-1.650	1.000
RSMG → RATL	0.000	0.000
RSMG → LDLPFC	-0.475	0.830
RSMG → LSMG	-1.037	1.000
RSMG → LATL	-0.699	1.000
RATL → RDLPFC	0.000	0.000
RATL → RSMG	0.000	0.000
RATL → RATL	-2.102	1.000
RATL → LDLPFC	0.000	0.000
RATL → LSMG	1.072	1.000
RATL → LATL	1.224	1.000
LDLPFC → RDLPFC	0.716	1.000
LDLPFC → RSMG	0.000	0.000
LDLPFC → RATL	0.000	0.000
LDLPFC → LDLPFC	-1.077	1.000
LDLPFC → LSMG	0.000	0.000
LDLPFC → LATL	0.000	0.000
LSMG → RDLPFC	0.000	0.000
LSMG → RSMG	0.308	0.673
LSMG → RATL	0.203	0.507
LSMG → LDLPFC	0.000	0.000
LSMG → LSMG	-0.803	0.848
LSMG → LATL	0.000	0.000
LATL → RDLPFC	-0.588	0.949
LATL → RSMG	0.000	0.000
LATL → RATL	0.931	1.000
LATL → LDLPFC	0.000	0.000
LATL → LSMG	0.539	0.853
LATL → LATL	-2.187	1.000

Note: Estimated modulatory effects under the pre-surprise condition for all bidirectional and self-connections

among six ROIs in the full-modulation DCM model: bilateral ATL, SMG, and dlPFC (L = left, R = right). The strength and direction of modulatory effects are indexed by posterior expectations (E_p); connections with posterior probability (P_p) > 95% are highlighted in bold.

We have now added this extended analysis to the Supplementary Information and revised the Discussion (p. 23) to clarify our modeling choices in the main text and to interpret the findings of data-driven model.

9. Limitations of Model Specification. For each hemisphere, the authors specified 15 DCM models. However, even with only three nodes, these 15 models do not exhaust all plausible configurations. For instance, the driving input could target nodes other than the ATL, and modulatory effects could exist on the connections between the ATL and dlPFC. These modeling choices likely reflect limitations inherent to Bayesian Model Selection (BMS), which requires a pre-defined and limited model space. While BMS is feasible for small models, it becomes impractical for larger networks due to the combinatorial explosion of possible configurations. A more suitable and scalable alternative would be the use of DCM Parametric Empirical Bayes (PEB), which allows efficient exploration of model space by pruning non-contributory parameters in a data-driven manner. Unfortunately, PEB was not implemented in this study.

Response:

We thank the reviewer for this comment. Our main analysis followed a deliberately hypothesis-driven strategy, using 15 carefully specified three-node models to test whether the SMG modulated either the ATL or the dlPFC. The ATL was retained as the entry node based on extensive theoretical and empirical support: as a core region for face identity processing and perceptual encoding, the ATL is well-positioned to receive bottom-up visual input in DCM models (Collins & Olson, 2014). This classical BMS framework is widely applied in confirmatory DCM studies with moderate model complexity (Friston et al., 2003; Rigoux et al., 2014; Huang et al., 2023; Heimhofer et al., 2024).

To address the reviewer's concern regarding the limitations of BMS and to complement our hypothesis-driven approach, we additionally conducted a PEB analysis on a fully connected six-node model (bilateral ATL, SMG, and dlPFC). The full model specification and results are reported in response to Comment 8 and in the Supplementary Information (Supplementary Fig. 3, Supplementary Table 3). Importantly, the PEB findings converged with the BMS results, showing a significant inhibitory modulation from the right SMG to the left ATL, but no evidence for SMG-dlPFC modulation.

10. Justification for Statistical Approach in DCM Analysis. In the "Dynamic causal modeling analysis" subsection, the authors state: "we extracted all the modulatory effect values of the selected model from 49 individuals and analyzed the modulatory effect on each inter-regional connection using one-sample t-test." However, it is unclear why Bayesian Model Averaging (BMA) was not used. BMA is typically preferred over relying on a single winning model, as it accounts for model uncertainty and provides more robust

parameter estimates across plausible models. The rationale for omitting BMA should be clarified.

Response:

We thank the reviewer for the comment. In the main hypothesis-driven analysis, we used BMS followed by classical one-sample t-tests on the winning model parameters, consistent with standard practice in hypothesis-driven confirmatory DCM studies (Rigoux et al., 2014; Huang et al., 2023; Heimhofer et al., 2024). To address concerns about model uncertainty, we additionally implemented Bayesian Model Averaging (BMA) within the extended PEB framework on the six-node model (see response to comment 8). The PEB-BMA results confirmed our main findings, again showing strong inhibitory modulation from the right SMG to the left ATL, with no significant effects from SMG to dlPFC. This convergence increases confidence in the robustness of our conclusions.

minor points

11. Caption of Table 1 can be more detailed: it would be helpful to remind the readers that there are no surprise trials in Run 1.

Response:

We thank the reviewer for this suggestion. To address this comment, in the revised manuscript, we have replaced Table 1 with Figure 2 (p. 7), which presents a more detailed bar-plot visualization of trial-level behavioral accuracy across all surprise trials during the fMRI experiment. We have updated the figure caption to explicitly note that no surprise trials occurred in Run 1, and that the first surprise trial appeared in Run 2.

Fig. 2. Behavioral results in the face-reporting task across surprise trials. Accuracy in the first surprise trial in Run 2 (dark orange bar) was 49%, significantly lower than in all subsequent control trials (light orange bars, all p s < 0.05, chi-square tests). Consistent with previous reports of the attribute amnesia effect^{12,20-22}, participants failed to report attended information during the first unexpected probe, but performance improved rapidly once memory demands were anticipated. Accuracies of each trial are displayed as ratios of correct/total participants, rounded to integers.

12. Software Description for DCM: The statement, “A DCM analysis of the effective connectivity between ROIs under the modulatory effects of the pre-surprise condition was conducted using SPM12 toolbox,” could be more accurately phrased as, “using the SPM software,” since multiple toolboxes are associated with SPM.

Response:

We thank the reviewer for this suggestion. In response, we have revised the relevant sentence of “A DCM analysis of the effective connectivity between ROIs under the modulatory effects of the pre-surprise condition was conducted using **SPM12 toolbox**,” in the Methods section to “*A DCM analysis of the effective connectivity between ROIs under the modulatory effects of the pre-surprise condition was conducted using **the SPM12**.*”

13. The term “exceedance probability” should be briefly explained when it first appears in the manuscript.

Response:

We thank the reviewer for this suggestion. In response, we have added a brief explanation of the term exceedance probability at its first appearance in the Results section (p. 13). Specifically, we now clarify that the exceedance probability is “*defined as the probability that a given model outperforms all others in explaining the observed data.*” We appreciate the reviewer’s attention to clarity and accessibility.

Reviewer #2 (Remarks to the Author):

In three experiments the authors aimed to demonstrate that attention and working memory are dissociative cognitive processes which are modulated by circuitry involving inhibitory attentional control by the supramarginal gyrus (SMG). In the first study involving task related fMRI they used an attribute amnesia task to identify brain regions involved in the regulating the transition of stimuli into working memory. This study showed that increased activation of the SMG was associated with inhibiting stimuli from entering working memory. This study also showed that there was a time course difference in BOLD signal activation in the bilateral anterior temporal lobe for face perception depending on whether attended to with or without memory. The following two experiments then used two different versions of a different cognitive task (change detection task) together with high-definition transcranial direct current stimulation (HD-tDCS) to show that stimulation of the SMG could reduce performance (slow reaction time) in the same identity/shape condition compared to the change identity/shape condition. This effect was replicated for anodal tDCS in both experiments. The authors interpreted this finding as direct causal evidence of a role of the SMG in attentional inhibition on the attended face/shape. Strengths of the studies presented included the task-related fMRI experiment showing a role of the SMG in uncoupling of attention and memory and the dynamic causal modelling

showing the activity in the functional network involved. A potential weakness were the HD-tDCS experiments and the assertion that these provide direct causal evidence that modulation of the SMG affected attentional inhibitory processes. The main weakness I see is the assertion from the behavioural results (reaction times) on the respective tasks supports a role of the SMG in these processes. Although the behavioural effect was replicated (which is a further study strength) both the task and small sized effect seen with the anodal conditions compared to sham are not convincing in supporting a specific role of the SMG in attention inhibition. An alternative task paradigm with a clearer behavioural effect (as seen with the attribute amnesia task with task-related fMRI) would have provided more convincing evidence. Also, a more focal form of brain stimulation (e.g., rTMS) may have been a better choice for demonstrating a causal role of this region in the effect. HD-tDCS involves stimulation (both excitatory and inhibitory) over a reasonably large cortical area which involves the SMG and several other neighbouring regions, so any specific behavioural effect (slowing of RT) in a particular condition could also potentially be attributed to modulation of other brain regions/networks other than the SMG.

General Response:

We sincerely thank the reviewer for the valuable comments. To address these concerns and strengthen the causal claim, we conducted an additional neuronavigated transcranial magnetic stimulation (TMS) experiment targeting the SMG. This experiment employed a theta-burst stimulation (TBS) protocol, which enables more spatially focal and functionally specific neuromodulation compared to HD-tDCS. The stimulation site was individually localized using structural MRI and neuronavigation, based on the SMG peak ([55, -44, 37]) identified in our original fMRI results.

Following methodological recommendations (e.g., Rossi et al., 2009; Kirkovski et al., 2023), participants were randomly assigned to one of three groups: continuous TBS (cTBS), intermittent TBS (iTBS), or sham. The cTBS and iTBS protocols were used to transiently suppress or enhance cortical excitability, respectively. All participants then completed the same change-detection task used in the original tDCS experiment. The results replicated and extended our original findings. Specifically, cTBS over the right SMG significantly enhanced inhibitory control over the residual identity representations, as indexed by slowed reaction times under the same-identity condition. These findings converge with our previous anodal HD-tDCS results and provides more spatially precise causal evidence supporting the inhibitory gating role of the SMG.

We have now integrated these new results into the revised manuscript as follows:

Results section (pp.19-21): A new subsection entitled “Causal modulation via neuronavigated TMS supports SMG’s inhibitory role” reports the behavioral outcomes under iTBS, cTBS, and sham groups, accompanied by “Fig. 8. Results of the TMS experiment”.

Fig. 8. Results of the TMS experiment. Ninety new participants were randomly assigned to one of three groups: intermittent theta-burst stimulation (iTBS), continuous theta-burst stimulation (cTBS) or sham stimulation. The bar graph displays reaction time results from the face-based change-detection task performed after stimulation for each group. $**p < 0.01$; $***p < 0.001$; ns = not significant.

Methods section (pp. 33-34): Detailed descriptions of the TMS procedures, stimulation parameters, neuronavigation protocol, and statistical analysis have been added:

“Neuronavigated TMS experiment. We conducted a neuronavigated transcranial magnetic stimulation (TMS) experiment to provide spatially precise causal evidence for the role of the SMG in inhibitory control. To enable precise targeting, high-resolution structural MRI images were acquired for each participant using a 3T Siemens Verio scanner (Germany) with a T1-weighted MPRAGE sequence (TR = 2300 ms, TE = 2.19 ms, flip angle = 9°, FOV = 256 × 256 mm², slice thickness = 1 mm, 176 slices). The stimulation target (MNI coordinates: x = 55, y = -44, z = 37) corresponded to the right SMG identified in our prior fMRI results. This location was individually localized using Visor 2.0 neuronavigation software (ANT Neuro, Netherlands) and tracked with a Polaris Spectra motion tracking system (NDI, Canada). Minor anatomical adjustments were made when necessary to account for individual variability. Magnetic stimulation was delivered via a figure-of-eight coil (outer diameter: 70 mm) connected to an NS5000 stimulator (YIRUIDE, Wuhan, China). Stimulation intensity was set to 80% of each participant’s resting motor threshold (RMT), defined as the minimum intensity required to elicit a motor-evoked potential of $\geq 50 \mu\text{V}$ in the left first dorsal interosseous muscle in at least 5 out of 10 trials during rest.

Participants were randomly assigned to one of three stimulation groups: active cTBS, active iTBS, or sham TBS^{68,70}. The cTBS protocol involved 600 pulses delivered continuously over 40 seconds (three pulses at 50 Hz every 200 ms). The iTBS protocol also comprised 600 pulses delivered over 200 seconds, consisting of 2-second trains (three pulses at 50 Hz every 200 ms), repeated every 10 seconds. For sham stimulation, the coil was positioned perpendicular (90°) to the scalp, preventing effective stimulation while preserving auditory and somatosensory sensations. Immediately following stimulation, participants completed the same change-detection task as used in the first tDCS experiment, designed to assess the inhibitory control over attended-without-memory face identity.”

Discussion section: In the Discussion, we interpret the TMS findings (p. 24): “Critically,

this inhibitory gating effect was further corroborated by a TMS experiment, which replicated and extended the tDCS findings with higher spatial specificity (Fig. 8). ”, and further address the limitation underlying the enhancement observed with cTBS rather than iTBS (p. 25): “Second, although cTBS is generally associated with reduced cortical excitability, our TMS experiment demonstrated that cTBS over the SMG produced increased functional effects—specifically, a significant enhancement of inhibitory control. This paradoxical facilitation is consistent with prior reports showing that cTBS can sometimes improve cognitive performance in a task- and region-specific manner^{68–70}. Such enhancement may be related to rebound excitability, state-dependent effects, or compensatory rebalancing within functional networks, underscoring the non-linear and context-dependent nature of stimulation effects in higher-order cortex^{68,71}.”

Additionally, regarding the suggestion to use an alternative behavioral task (e.g., attribute amnesia), we agree that the attribute amnesia task yielded clearer group-level fMRI effects. However, its behavioral sensitivity to experimental manipulation is limited, as it relies on a single surprise trial. In contrast, the change-detection task enabled us to collect sufficient within-subject trials to detect modulation effects of neuromodulation, which was critical for the HD-tDCS and TMS studies. We have clarified this rationale in the Results section. Specifically, we added the following sentence:

“In the first tDCS experiment, we used a modified change-detection paradigm with colored faces as stimuli (Fig. 6c). Compared with the attribute amnesia task, which relies on a single surprise trial, the change-detection task provided sufficient within-subject trials, making it more sensitive for detecting stimulation effects in the neuromodulation experiments.”

We are grateful to the reviewer for prompting these important additions. We believe that the inclusion of the TMS experiment enhances the scientific rigor and interpretive strength of our study and directly addresses concerns about causal inference and spatial specificity.

I have the following queries/comments to assist with further improving the paper.

1. Results. Table 1. This data would be better presented as bar graph showing SD. Why wasn't a RMANOVA conducted to examine these behavioural results?

Response:

We thank the reviewer for this comment. In line with your recommendation, we have revised the manuscript to present the data previously shown in Table 1 as a bar graph (now Fig. 2).

Fig. 2. Behavioral results in the face-reporting task across surprise trials. Accuracy in the first surprise trial in Run 2 (dark orange bar) was 49%, significantly lower than in all subsequent control trials (light orange bars, all p s < 0.05, chi-square tests). Consistent with previous reports of the attribute amnesia effect^{12,20–22}, participants failed to report attended information during the first unexpected probe, but performance improved rapidly once memory demands were anticipated. Accuracies of each trial are displayed as ratios of correct/total participants, rounded to integers.

Regarding the statistical approach, we respectfully note that the dependent variable—trial-level accuracy (correct vs. incorrect)—was binary in nature and therefore does not yield SD. Moreover, RMANOVA is not an appropriate method for this type of categorical data. Instead, consistent with prior studies employing the attribute amnesia paradigm (e.g., Chen et al., 2015, 2016, 2019; Tam et al., 2021; Fu et al., 2024), we applied chi-square tests to compare categorical accuracy data across discrete surprise trials. Specifically, to assess the attribute amnesia effect, we tested whether accuracy on the first surprise trial—where memory for identity was unexpectedly probed—was significantly lower than on subsequent control surprise trials. This analytical strategy follows established practices in attribute amnesia research and provides a valid test of the core behavioral signature of the effect.

2. Results. Figure 2. Instead of using blue, the figures would be clearer showing the increased activation in the memory regions as red.

Response:

We thank the reviewer for the comment. The current figure visualizes the statistical contrast “pre-surprise > post-surprise,” with regions showing greater activation in the pre-surprise condition—most notably the supramarginal gyrus (SMG)—displayed in warm colors (e.g., red/yellow), and regions with greater activation in the post-surprise condition, including memory-related areas, shown in cool colors (e.g., blue/green). This color scheme follows standard neuroimaging conventions, where warm and cool colors intuitively represent positive and negative contrast values, respectively, and helps avoid misinterpretation of contrast direction.

To improve clarity, we have revised the figure caption to explicitly state the contrast

direction and associated task conditions.

3. Methods. tDCS experiments 1 and 2. Please clarify whether these studies were randomised.

Response:

We thank the reviewer for this suggestion. In both experiments, participants were randomly assigned to one of three stimulation conditions: anodal, cathodal, or sham. This randomization procedure was consistent with prior studies employing comparable stimulation protocols (e.g., Naros et al., 2016; Rossi et al., 2009; Kirkovski et al., 2023).

We have now clarified this in the Methods section with the following sentence:

“For the tDCS experiments, participants were randomly assigned to the anodal, cathodal, or sham stimulation groups^{72,74,78}.”

4. Methods. tDCS experiments. Was a power analysis conducted for these experiments? tDCS behavioural effects are very small (Dedoncker et al., 2016, *Brain Stimul*)

Response:

We thank the reviewer for this comment. To ensure sufficient power for detecting behavioral effects in our design, we conducted an a priori power analysis using G*Power 3.1 (Faul et al., 2007). Based on pilot data collected using change-detection task, the expected effect size for the change-distracting effect was estimated at $d_z = 0.67$ (Fu et al., 2021). The analysis indicated that a sample size of 20 participants would provide 80% power to detect this effect at an alpha level of 0.05.

However, given prior evidence that tDCS typically yields smaller behavioral effects (Dedoncker et al., 2016, *Brain Stimul*), we conservatively increased the sample size to 30 participants per group in both tDCS experiments to ensure adequate statistical sensitivity. This sampling rationale and power calculation have now been explicitly added to the revised Methods section (Participants subsection, p. 26).

5. Results. tDCS experiments. Please provide the results from the full model ANOVA, including specific main effects and interaction effects.

Response:

We thank the reviewer for this comment. In response, we now report the results of the full model ANOVA.

For **the first tDCS experiment**, a 3 (Stimulation: Anodal, Cathodal, Sham) \times 2 (Condition: Same-identity, Change-identity) mixed-design ANOVA revealed a trend-level interaction between stimulation and condition ($F(2,87) = 2.980$, $p = 0.056$, $\eta_p^2 = 0.064$), with no significant main effects of stimulation or condition. Although the interaction did not reach significance, the pattern suggested potential polarity-dependent effects, raising the possibility that combining both active groups into a single analysis may have masked

polarity-specific modulation (Antal et al.,2017; Weller et al.,2020). Based on previous literature, the polarity-specific effects are typically examined by directly comparing each active group with sham (Pope et al.,2012; Antal et al.,2017; Weller et al.,2020; Cheng et al.,2022). Therefore, we conducted planned 2 (Group: Active vs. Sham) × 2 (Condition) ANOVAs separately for the anodal and cathodal groups. The results showed that only the anodal group exhibited a significant interaction with sham ($F(1,58) = 6.465, p = 0.014, \eta^2 = 0.100$), with longer RTs in the same-identity than in the change-identity condition, indicating stronger inhibition of residual face-identity traces. The cathodal group showed no such effect.

The second tDCS experiment replicated this pattern: the full 3×2 ANOVA again showed only a trend-level interaction ($F(2,87) = 2.659, p = 0.076, \eta^2 = 0.058$), but planned comparisons confirmed a significant interaction for anodal vs. sham ($F(1,58) = 4.518, p = 0.038, \eta^2 = 0.072$), with no effect for cathodal stimulation. These findings closely mirror the first experiment, demonstrating polarity-dependent enhancement of inhibitory control under anodal tDCS.

In addition, we have added **a new TMS experiment** according to your suggestion, which provided stronger causal evidence with higher spatial specificity. Here, the full 3×2 ANOVA revealed a significant interaction ($F(2,87) = 5.586, p = 0.005, \eta^2 = 0.114$). Planned contrasts again showed polarity-specific effects: cTBS significantly enhanced inhibition relative to sham ($F(1,58) = 8.211, p = 0.006, \eta^2 = 0.124$), while iTBS showed no effect. This polarity-selective modulation aligns with the tDCS findings and strengthens the evidence for our conclusion.

These results have now been fully incorporated into the revised Results section of the main text for both the tDCS and TMS experiments (pp. 16, 18, 20).

6. Results. tDCS experiments. Note that reporting the instances of side effects for the different stimulation conditions is not the same as directly testing adequacy of participant blinding, e.g., asking participants whether they thought that they received active or sham tDCS. If blinding was not assessed, then this should be stated as a limitation.

Response:

We thank the reviewer for this comment. In addition to administering the Questionnaire of Sensations Related to Transcranial Electrical Stimulation (tES), we also asked participants—at the end of each tDCS session—to indicate whether they believed they had received active or sham stimulation. Participants' responses were approximately evenly distributed across stimulation conditions, with no clear bias toward active or sham guesses, suggesting that they were largely unable to distinguish between stimulation types.

We have updated the Methods section to clearly distinguish between sensation reporting and blinding assessment, and added the following sentence:

"In addition, participants were asked to guess the type of stimulation they had received

(active vs. sham), and their responses were approximately evenly distributed across conditions, suggesting effective blinding."

Reviewer #3 (Remarks to the Author):

This is a timely study on the interaction of attentive selection and working memory, the methods are sound and the paper generally well written. I do, however, have one concern that relates to the way the authors present their findings where they stress the uniqueness of their results. First, other than stated in the first sentence of the abstracts, although everyone in the field would agree that both processes are closely related, no-one would claim that they are inseparable. Likewise, their study is not the first to provide evidence that the two processes are dissociable in terms of brain physiology (e.g. McNab & Klingberg, 2008). Hence, I recommend the authors to formulate their statements with more care.

Response:

We sincerely thank the reviewer for highlighting the methodological strengths of our study and for the constructive feedback on how to frame our contribution more carefully.

In line with this suggestion, we have revised the Abstract, Introduction, and Discussion to present our findings with greater precision and balance. Prior work, such as McNab and Klingberg (2008), has indeed provided important evidence that attentional filtering acts as a gatekeeper for WM and demonstrated overlapping neural mechanisms between attentional selection and WM encoding. Our study builds on this foundation but further suggests that attentional selection and WM encoding are not only dissociable, but can also be governed by a novel inhibitory mechanism: even when a stimulus has been fully attended, its representation may still be actively suppressed to prevent WM encoding.

In the revision, we made it clear that our focus lies specifically on the attentional selection of external information and its encoding into WM (rather than on the general relationship between attention and WM). We also emphasized how our findings extend prior work, thereby providing greater precision and avoiding ambiguity. Moreover, to ensure clarity, we have refined the phrasing of key claims. For instance, the first sentence of the Abstract has been changed from "*Attention and WM ... considered inseparable processes ...*" to "*Attention and WM ... considered closely linked processes...*" Similarly, we have replaced potentially overstated phrases such as "*the first study to...*" with more accurate formulations like "*we provide converging neural and causal evidence...*".

We are grateful for this valuable suggestion, which has helped us sharpen our conceptual framing and highlight the genuine novelty of our contribution without overstating its uniqueness.

REVIEWERS' COMMENTS

Reviewer #1:

The authors have addressed most of my comments. I have one additional suggestion: While the revisions to the Participants section are helpful, the authors should also acknowledge the demographic characteristics of their sample as a limitation. In particular, the restricted age range and educational homogeneity (i.e., university students in Guangzhou, China) may limit the generalizability of the findings to broader and more diverse populations. Explicitly noting this in the Discussion as a study limitation would further strengthen the manuscript's transparency and contextualization.

Response

Thank you for this suggestion. We have added this limitation to the Discussion to strengthen the manuscript's transparency and contextualization. Specifically, we now acknowledge that our sample—predominantly young adult university students from Guangzhou, China—has a restricted age range and educational homogeneity, which may limit generalizability to broader populations.

The following sentences have been added to the Discussion:

“We have added these sentences in the Discussion “Finally, our sample consisted primarily of young adult university students from Guangzhou, China, yielding a restricted age range and relatively homogeneous educational background; therefore, generalizability to other age groups, educational background, and cultural contexts may be limited. Future work should test whether SMG-mediated inhibitory gating generalizes across more diverse populations.”

Reviewer #2:

The addition of the new rTMS experiment significantly strengthens the authors' conclusions. My other concerns and queries have been adequately addressed.

Response

We are delighted to learn that the reviewer finds the addition of the rTMS experiment to be a significant improvement to our study. Their constructive feedback throughout the review process has been invaluable in shaping the final manuscript.